# Exploring plant-derived phytochrome chaperone proteins for light-switchable transcriptional regulation in mammals

Deqiang Kong [1,4], Yang Zhou [1,2,4], Yu Wei[1,4], Xinyi Wang [1], Qin Huang[1], Xianyun Gao[1], Hang Wan[1], Mengyao Liu[1], Liping Kang[1], Guiling Yu[1], Jianli Yin[1,3], Ningzi Guan [1] ✉ & Haifeng Ye [1,2] ✉

Synthetic biology applications require finely tuned gene expression, often mediated by synthetic transcription factors (sTFs) compatible with the human genome and transcriptional regulation mechanisms. While various DNA-binding and activation domains have been developed for different applications, advanced artificially controllable sTFs with improved regulatory capabilities are required for increasingly sophisticated applications. Here, in mammalian cells and mice, we validate the transactivator function and homo-/ heterodimerization activity of the plant-derived phytochrome chaperone proteins, FHY1 and FHL. Our results demonstrate that FHY1/FHL form a photosensing transcriptional regulation complex (PTRC) through interaction with the phytochrome, ΔPhyA, that can toggle between active and inactive states through exposure to red or far-red light, respectively. Exploiting this capability, we develop a light-switchable platform that allows for orthogonal, modular, and tunable control of gene transcription, and incorporate it into a PTRC-controlled CRISPRa system (PTRC$_{dcas}$) to modulate endogenous gene expression. We then integrate the PTRC with small molecule- or blue light-inducible regulatory modules to construct a variety of highly tunable systems that allow rapid and reversible control of transcriptional regulation in vitro and in vivo. Validation and deployment of these plant-derived phytochrome chaperone proteins in a PTRC platform have produced a versatile, powerful tool for advanced research and biomedical engineering applications.

The transcription of protein-coding genes is finely orchestrated through the coordinated interplay of transcription factors (TFs), which bind DNA in a sequence-specific manner, and various chromatin-associated factors that modulate chromatin structure, all in conjunction with RNA polymerase II[1]. Most TFs are comprised of a well-defined DNA-binding domain (DBD) and a separate activation domain (AD), also referred to as a transactivator[2]. To achieve highly sensitive and orthogonal regulation of gene expression in mammals, synthetic transcription factors (sTFs) have been designed through the modular assembly of DBDs and ADs from bacterial, fungal (especially yeast),

[1]Shanghai Frontiers Science Center of Genome Editing and Cell Therapy, Biomedical Synthetic Biology Research Center, Shanghai Key Laboratory of Regulatory Biology, Institute of Biomedical Sciences and School of Life Sciences, East China Normal University, Dongchuan Road 500, Shanghai 200241, China. [2]Wuhu Hospital, Health Science Center, East China Normal University, Middle Jiuhua Road 263, Wuhu City, China. [3]Chongqing Key Laboratory of Precision Optics, Chongqing Institute of East China Normal University, Chongqing 401120, China. [4]These authors contributed equally: Deqiang Kong, Yang Zhou, Yu Wei. ✉e-mail: nzguan@bio.ecnu.edu.cn; hfye@bio.ecnu.edu.cn

insect, or viral sources[3–5]. For example, virion protein 16 (VP16), and its four-repeat variant, VP64, from herpes simplex virus type 1[5,6] are potent transactivators that function in a wide range of cell types, which has led to their broad adoption in sTFs. In addition, these ADs can be incorporated into chimeric fusions, such as the hybrid tripartite activator VPR[4,7] (consisting of VP64, human NF-κB trans-activating subunit p65[8], and Epstein-Barr Virus Rta[9]) and p65-HSF[4,7] (combining human NF-κB trans-activating subunit p65 with the activation domain from human heat-shock factor 1[10]).

The development of controllable sTFs that enable precise control of gene expression is currently a major objective in precision medicine research. In particular, several small molecule- or blue light-regulated sTFs have been designed by combining ADs with small molecule- or blue light-responsive DBDs that bind or dissociate from promoters containing the specific cognate DNA sequence upon exposure (or withdrawal of) their respective triggers, thereby regulating the transcription of target genes[11–14]. However, the control mechanisms of these synthetic transcription factors (sTFs) are intrinsically linked to their specific DNA-binding domains (DBDs), which are often inflexible. For instance, in tetracycline-controlled sTFs, the DBD is specifically confined to the tetracycline-responsive protein TetR[15]. Changing the DBD to a non-TetR variant removes the tetracycline-mediated control, thus reducing their adaptability and compatibility with other systems. To overcome these limitations of DBDs and achieve precise control of transcriptional regulation, proteins that exhibit inducible dimerization have been adopted in the design of controllable TFs. In such sTFs, the DBD and AD are respectively fused with two proteins that mutually interact upon exposure to an inducer, such as blue light or a small molecule ligand, resulting in the AD entering close proximity with the promoter region to assemble with the DBD and consequently form a complete sTF[16–24]. However, low dimerization efficiency and/or excessive distance between the AD and promoter often limit the transactivation function of such dimerization-dependent sTFs. Thus, despite these considerable advances, there remains significant demand for high-efficiency, tightly controllable TFs that exhibit robust transcriptional activation and compatibility across diverse regulation systems, without the limitations of current DBDs.

Plants represent a vast genetic resource that has remained untapped in mammalian synthetic transcriptional regulation systems. In plants, phytochromes (PhyA - PhyE)[25] regulate various light responses through interactions with their respective chaperone proteins, such as phytochrome interacting factor 3 (PIF3, which interacts with PhyB), far-red elongated hypocotyl1 (FHY1, which interacts with PhyA), and FHY1-LIKE (FHL, which also interacts with PhyA). PIF3 is well established as a transcription factor (TF) in plants[26]. Given their dual functionality as light-responsive proteins and transcriptional activation elements, proteins like PIF3 present a compelling opportunity to be employed as transactivators in designing light-responsive synthetic transcription factors (sTFs) for mammalian cells. However, there is a notable shortage of plant-derived TFs effectively utilized in regulating transcription in mammalian systems[27].

Here, we demonstrate that two plant-derived phytochrome chaperone proteins, FHY1 and FHL, can function as transcriptional activators in mammalian cells, and identify the essential DNA sequences in their respective functional domains. Further, our evidence illustrates their capacity for homodimerization as well as heterodimerization with each other, leading to our development of an FHL/FHY1-mediated split-Cre recombinase system for targeted DNA recombination in both mammalian cells and mice. Moreover, exposure to red light induces ΔPhyA binding to FHL/FHY1 in the presence of the chromophore, PCB, resulting in a photosensing transcriptional regulation complex (PTRC) that suppresses their transcriptional activation function. Upon exposure to far-red light, ΔPhyA dissociates from FHL/FHY1, disassembling the PTRC and disinhibiting FHL/FHY1 transcriptional activation of target genes.

Based on this inducibility, we constructed a light-controlled system for transcriptional regulation of endogenous genes in mammalian cells and mice using a PTRC-mediated CRISPRa system (PTRC$_{dcas}$). In addition, our results show that PTRC is compatible with several transcriptional regulation systems controlled by either small-molecular triggers or blue light, enabling alternative activation modes with red light-inducible transcriptional suppression. These versatile transactivators expand the pool of sources for mammalian-compatible sTFs to include plants. By circumventing the constraints of conventional synthetic biology approaches to transcriptional regulation that rely on control of sTF-DNA interactions or modulating sTF integrity, this PTRC platform can be broadly applied in the development of controllable sTFs for a wide range of specialized transcriptional regulation needs in research and biomedical applications.

## Results
### Validation of plant-derived phytochrome chaperone proteins for transcriptional activation in mammalian cells
To develop light-responsive transactivator modules, we selected three *Arabidopsis*-derived phytochrome chaperone proteins, PIF3, FHY1, and FHL, based on their well-established role in light-dependent processes in plants[28,29]. To assess whether these proteins could exhibit transcriptional regulatory activity in mammalian cells, we engineered a panel of sTFs by individually fusing PIF3, FHY1, and FHL with a panel of DBDs, including from yeast Gal4[30,31], *X species* Tet repressor proteins (TetR)[32], the BR60 repressor of 3-chlorobenzoate catabolism (CbaR)[33] from *Comamonas testosteroni*, or the vanillic acid-dependent repressor (VanR)[34] from the VanAB gene cluster of *Caulobacter crescentus*. The reporter constructs contained a corresponding operator sequence upstream of a minimal immediate early promoter from human cytomegalovirus (CMV), i.e., 5×UAS-P$_{hCMVmin}$, 7×TetO-P$_{hCMVmin}$, 2×CbaO-P$_{hCMVmin}$, or 2×VanO-P$_{hCMVmin}$, which drove the expression of secreted human placental alkaline phosphatase (SEAP) (Fig. 1a and Supplementary Fig. 1a). We then transfected the plasmids encoding respective sTF and reporter into HEK293T cells and detected SEAP production in culture supernatants 24 hours after transfection. FHY1 exhibited activation of SEAP expression that was comparable to, or slightly lower than, that induced by the positive control transactivator VP16, derived from herpes simplex virus[5]. In contrast, FHL, when fused with TetR, CbaR, and VanR, showed significantly enhanced activation of SEAP expression compared to VP16. However, HEK293T cells expressing PIF3 demonstrated negligible transcriptional activity (Fig. 1b, c and Supplementary Fig. 1b, c).

To confirm that the sTF indeed controlled SEAP production, we fused FHL and FHY1 with the resveratrol (RES)-responsive transcriptional repressor of the TtgABC efflux pump, TtgR, in *Pseudomonas putida*[11], or the protocatechuic acid (PCA) responsive transcriptional repressor, PcaV, from *Streptomyces coelicolor*[12,35]. In these assays, the sTFs should bind to their corresponding promoters (O$_{TRC1}$-P$_{hCMVmin}$; P$_{5×PcaV}$, 5×PcaA-P$_{hCMVmin}$) in the absence of small-molecule triggers, and subsequently recruit the transcription initiation complex to induce target gene expression. Alternatively, in the presence of small-molecule triggers, the sTFs should be released from their chimeric promoters, leading to suppression of target gene expression (see schematic for RES-induced transcriptional activation in Fig. 1d and PCA-induced transcriptional activation in Supplementary Fig. 1d). Detection of SEAP production revealed that treatment with small-molecule triggers, RES or PCA, resulted in dose-dependent inhibition of SEAP production (Fig. 1e, f and Supplementary Fig. 1e, f), indicating that FHL/FHY1 could activate target gene expression regardless of which DBD was used. Moreover, FHL and FHY1 could also mediate the expression of other proteins of interest, including luciferase and EGFP (Supplementary Fig. 2a, b). To assess the FHL and FHY1 compatibility with different cell types, we introduced the sTF-Gal4 fusions and reporter into several commonly used mammalian cell lines. Microplate

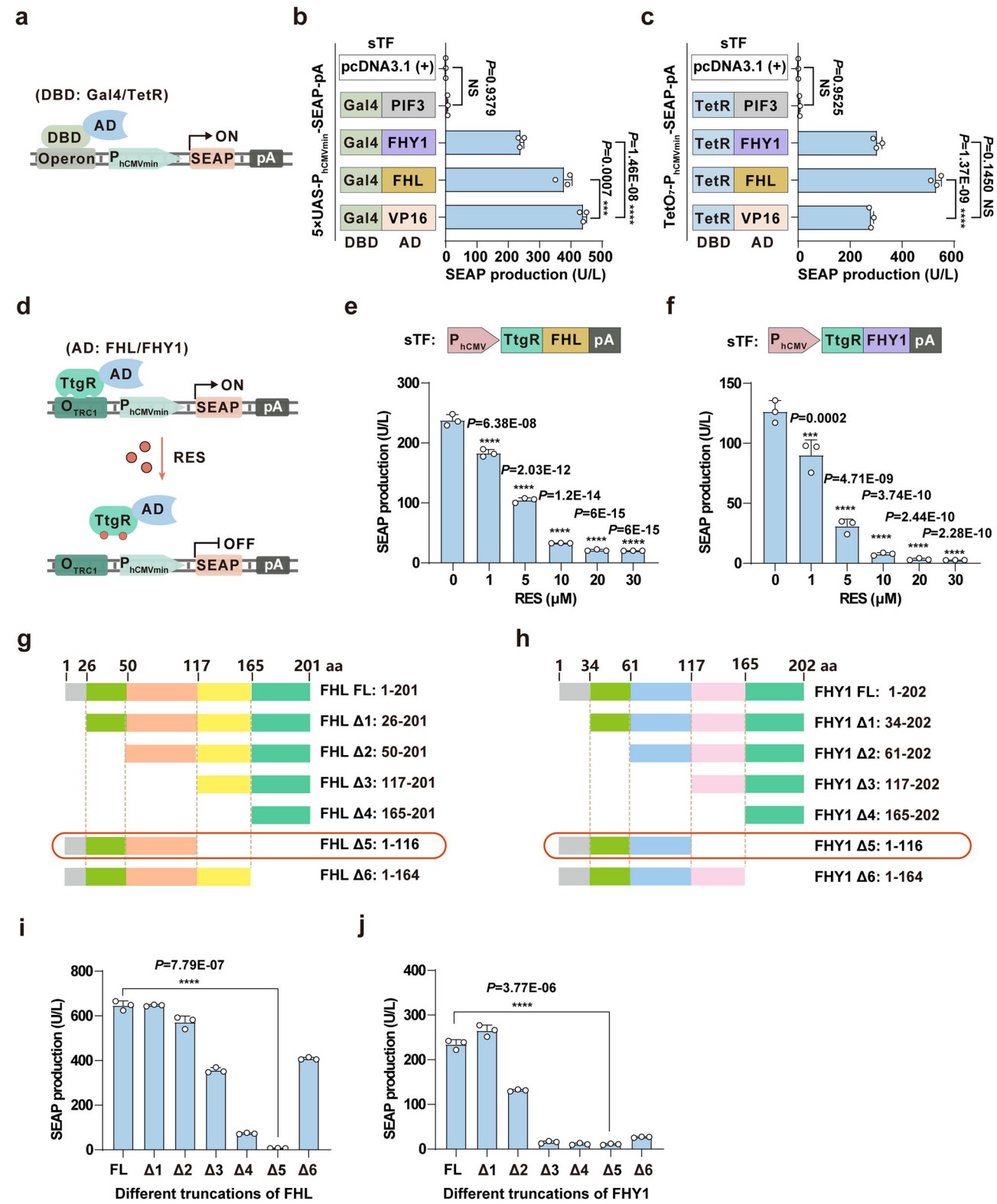

reader assays indicated that SEAP could be detected in all cell lines at significantly higher levels than reporter controls (Supplementary Fig. 3). These results suggested that FHY1 and FHL could potentially serve as robust and biocompatible transactivators for various possible applications in mammalian cells.

To determine which functional domains of FHL and FHY1 were required for their transcriptional regulatory function, we conducted sequence alignments to identify potentially conserved functional domains between the two proteins. This alignment revealed that homology between FHL and FHY1 was limited primarily to the nuclear localization signal (NLS), nuclear export signal (NES), and extreme C-terminus regions (Supplementary Fig. 4). To evaluate the impacts of these conserved protein domains on transcriptional activation function, we subsequently fused different truncation variants of FHL and FHY1 to the Gal4 domain (Fig. 1g, h). We observed that deletion of either the NLS/NES or extreme C-terminal regions from FHY1, but not

**Fig. 1 | Validation of the transcriptional activation functions of *Arabidopsis*-derived phytochrome chaperone proteins in mammalian cells. a** Schematic of the design for screening potential transactivators. The DNA binding domain (DBD: Gal4 or TetR) is fused with candidate transactivators to create synthetic TF (sTF), which translocates into the nucleus and binds to the synthetic specific promoter to initiate target gene expression. **b, c** Quantification of gene expression mediated by Gal4-based (**b**) and TetR-based (**c**) sTFs. **d** Schematic representation of a resveratrol (RES)-triggered transcriptional inhibition system using FHL/FHY1. Without RES, the sTF (TtgR-FHL/FHY1) binds to a chimeric $P_{hCMVmin}$ promoter, with a specific DBD binding site positioned upstream of the promoter to initiate target gene expression. In the presence of RES, TtgR-FHL/FHY1 is released from the chimeric promoter, terminating target gene expression. **e, f** Dose-dependent gene expression dynamics mediated by a RES-responsive sTF. HEK293T cells transfected with TtgR-FHL (**e**) or TtgR-FHY1 (**f**), along with the SEAP reporter ($O_{TRC1}$-$P_{hCMVmin}$-SEAP-pA),

were cultivated with different amounts of RES for 24 hours, before measuring SEAP production. **g, h** Schematic of different truncation variants of FHL (**g**) and FHY1 (**h**), with different regions differentiated by color. The amino acid positions of the truncation sites are indicated at the top, and the amino acid regions for each truncation of FHL/FHY1 are detailed at the bottom. FL denotes full length. Mutants highlighted with a red box are referred to as ΔFHL/ΔFHY1 in subsequent text. **i, j** Quantification of gene expression mediated by different FHL/FHY1 truncation variants. HEK293T cells transfected with reporter (5×UAS-$P_{hCMVmin}$-SEAP-pA) and different truncation variants of FHL (**i**) or FHY1 (**j**) fused to Gal4 were cultured for 24 hours, before measuring SEAP production. Data in **b, c, e, f, i, j** are presented as means ± s.d.; $n = 3$ independent experiments, with three technical replicates for each. Statistical comparisons in **b, c, e, f** were performed by one-way ANOVA, in **i, j** were performed by two-tailed Student's $t$ test; NS, not significant, ***$p < 0.001$, ****$p < 0.0001$. Source data are provided as a Source Data file.

its extreme N-terminus, led to a significant decrease in SEAP production (Fig. 1i), but had relatively little effect on FHL function (Fig. 1j). These results suggested that amino acids 117-165 in FHL, between the NLS/NES and C-terminus, likely contained the transactivator domain responsible for transcriptional regulatory activity. In contrast, FHY1 activity appeared to require all regions except for its extreme N-terminus. Notably, fusion of the Gal4 or TetR DBDs with the C-terminal regions of full-length FHL or FHY1 significantly inhibited their transcriptional regulation of SEAP production (Supplementary Fig. 5). These results indicated that specific phytochrome chaperone proteins, which are known to be involved in plant transcriptional regulation, could serve as transactivators to control reporter gene expression in mammalian cells.

## An FHL/FHY1-based split-Cre recombinase system for DNA recombination

Since protein-protein interactions can be engineered for a wide range of experimental purposes, and both FHL and FHY1 have been previously shown to undergo homo- and/or hetero-dimerization alone or with each other in *Arabidopsis* [36], we next investigated whether FHL and FHY1 could also dimerize in various combinations in HEK293T cells. To this end, we constructed a split-Cre recombinase system activated by homo- or heterodimerization of FHY1 or FHL. In this system, two fragments of Cre recombinase (CreN: residues 1–59; CreC: residues 60–343) were individually fused to FHL or FHY1 (Fig. 2a). Upon FHL or FHY1 dimerization, the catalytic activity of Cre recombinase is restored, leading to excision of *loxP*-flanked STOP DNA sequences and subsequent gene expression. SEAP production could be detected both by the homodimerization of FHL or FHY1 mediated the restoration of split-cre recombinase and the heterodimerization of FHL and FHY1 mediated the restoration, demonstrating that FHL and FHY1 could form homo- and heterodimers in mammalian cells (Supplementary Fig. 6).

We then used this split-Cre recombinase system to identify the functional regions of FHL and FHY1 responsible for dimerization. We generated a set of FHY1 and FHL deletion mutants lacking either the N- or C-terminus regions (Fig. 1g, h) to examine their potential effects on dimerization. We fused each truncated form to different fragments of split-Cre. SEAP production demonstrated that FHL/FHY1 dimerization-dependent restoration of split-Cre recombinase activity was severely compromised in mutants lacking the extreme N-terminus of either protein, while further deletion of the NLS and NES regions resulted in almost wholly abolished Cre catalytic activity (Supplementary Fig. 7). By contrast, deleting the C-terminus (117–201 aa) resulted in significantly increased split-Cre recombinase activity through dimerization of the ΔFHL and/or ΔFHY1 variants (denoted as FHL Δ5/FHY1 Δ5 in Fig. 1g, h), reaching three to five-fold higher activity than that detected with full-length protein. Notably, minimal SEAP production was observed when either CreN or CreC fused with various ΔFHL/ΔFHY1 variants (Fig. 2b). These results showed that the N-terminus of FHL/

FHY1 was required for dimerization, whereas the C-terminus could potentially hinder dimerization. We next compared the hetero-dimerization efficiency of ΔFHL/ΔFHY1 with that of DocS/Coh2, a protein pair commonly used for spontaneous dimerization assays [37], using the split-Cre recombinase system, and found that ΔFHL/ΔFHY1 could restore higher split-Cre recombinase activity than DocS/Coh2 (Fig. 2c, d).

Based on our above in vitro evidence in HEK293T cells, we then tested ΔFHL/ΔFHY1 dimerization in mice in vivo. For these experiments, plasmids encoding the ΔFHL/ΔFHY1 (CreN-ΔFHL and ΔFHY1-CreC) or DocS/Coh2 (CreN-Coh2 and DocS-CreC) inducible split-Cre recombinase systems with a luciferase reporter, were delivered to the mouse liver by hydrodynamic injection of the tail vein (Fig. 2e). At 24 hours post-injection, luciferase expression levels were significantly higher in mice expressing the ΔFHL/ΔFHY1 dimer-dependent system compared to those with the DocS/Coh2-inducible system (Fig. 2f, g).

To evaluate the functionality of ΔFHL/ΔFHY1-split-Cre in DNA recombination at a genomic locus, we introduced the ΔFHL/ΔFHY1-dependent split-Cre system into tdTomato transgenic mice (Gt (ROSA) 26Sor^tm14(CAG-tdTomato)Hze), which carry a Cre reporter allele designed with a *loxP*-flanked STOP cassette that prevents CAG promoter-driven transcription of tdTomato. Plasmids encoding the ΔFHL/ΔFHY1- or DocS/Coh2-inducible split-Cre system were delivered to the livers of Cre-tdTomato mice by hydrodynamic tail vain injection (Fig. 2h). At seven days post-injection, IVIS quantification of tdTomato signal in isolated livers showed that signal intensity was significantly greater in livers of mice expressing the ΔFHL/ΔFHY1-dependent split-Cre system compared to mice injected with the DocS/Coh2-based system (Fig. 2i, j). Moreover, qPCR and Western blot analyses of the isolated liver tissues verified the successful induction of split-Cre recombinase by ΔFHL/ΔFHY1 (Fig. 2k, l), thereby demonstrating that this system could be used for inducible genome editing in mice. Fluorescence microscopy of liver sections (Fig. 2m) further validated our results. Collectively, these results demonstrated that spontaneous hetero- or homo-dimerization of FHL and/or FHY1 could regulate the expression of split proteins in mammalian cells in vitro and in vivo.

## A PTRC-mediated transcriptional regulation system

After identifying functional domains in FHL and FHY1 required for transcriptional activation partially overlaps with the regions reported to mediate interaction with PhyA under red light illumination [28], we next examined whether transcriptional activation and deactivation could be controlled by incorporating a ΔPhyA (the truncated phytochrome A: 1–617 aa) into the FHL-mediated transcriptional activation system. In this system, sTF Gal4-FHL should bind to its synthetic promoter to initiate the expression of the gene of interest (e.g., SEAP) under dark conditions. Upon exposure to red light, ΔPhyA protein specifically binds to FHL in the presence of the chromophore, PCB, forming a photosensing transcriptional regulation complex (PTRC, Fig. 3a), consequently suppressing FHL transcriptional regulatory

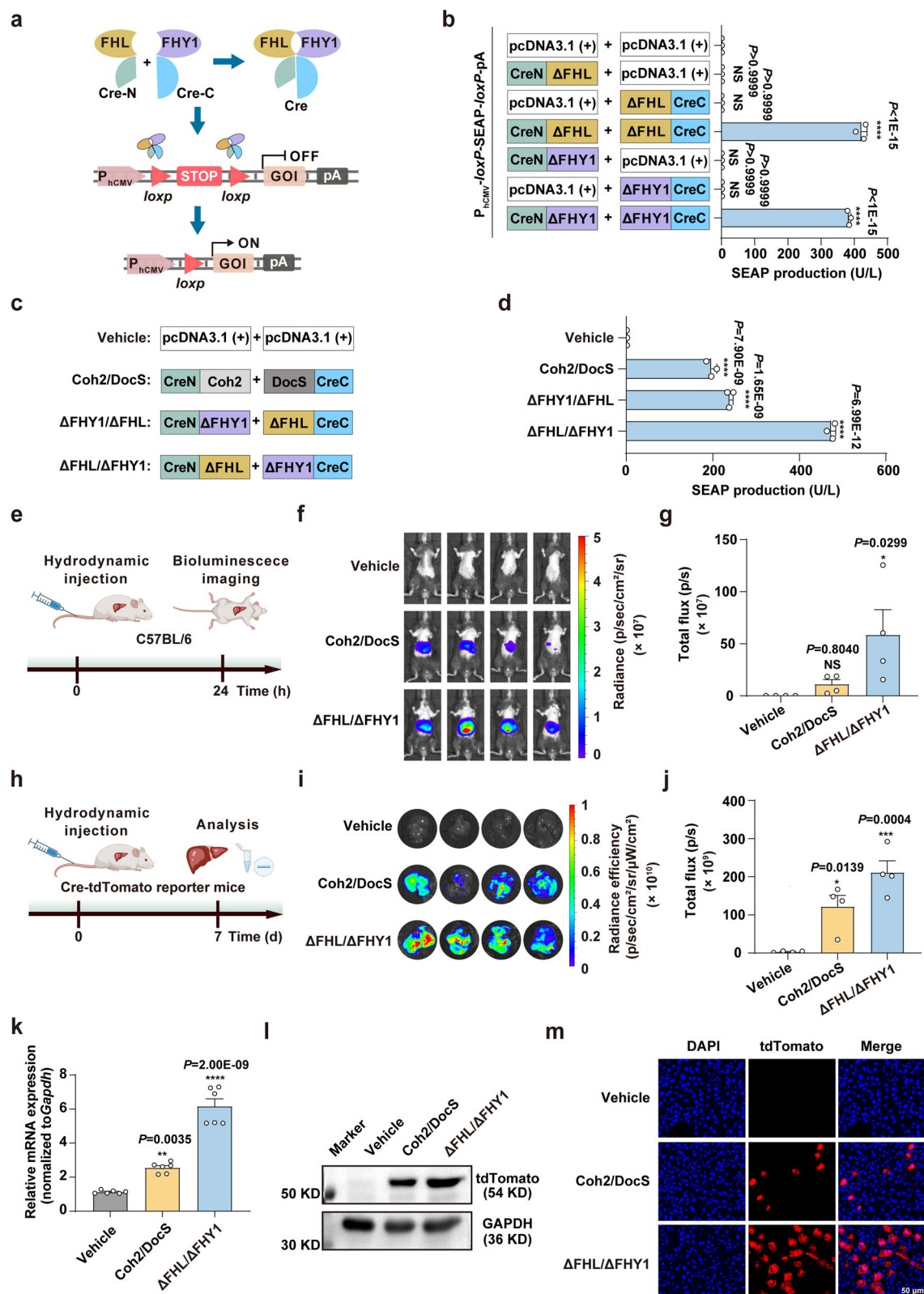

activity. Subsequent illumination with far-red light (730 nm) induces ΔPhyA dissociation from FHL, resulting in PTRC disassembly and restoration of target gene expression (Fig. 3b).

To optimize PTRC induction in HEK293T cells, using SEAP as a reporter gene, we tested the performance of the light-responsive transcriptional repressor in different configurations. Fusion of ΔPhyA with 2×ZIM3 (zinc finger imprinted 3)[38] resulted in an approximately

4-fold decrease in SEAP production upon exposure to red light (660 nm, 1 mW/cm², 24 hours) compared to that in non-illuminated cells (Supplementary Fig. 8a). Gene expression could be repressed to a greater extent by increasing the number of plasmids encoding ΔPhyA-2×ZIM3 (Supplementary Fig. 8b). Moreover, we noted that the repression efficiency of the light-responsive repressor decreased as the sTF copy number increased (Fig. 3c and Supplementary Fig. 9a).

**Fig. 2 | Design and characterization of a split-Cre recombinase system for DNA recombination based on FHL/FHY1 dimerization in mammalian cells and mice.**
**a** Schematic of an FHL/FHY1 dimerization-dependent split-Cre recombinase system. Cre recombinase was split into Cre-N (residues 1–59 aa) and Cre-C (residues 60–343 aa) fragments and fused to one of the proteins in the FHL/FHY1 pair. The split-Cre recombinase catalytic activity is restored when the two Cre fragments enter close proximity following heterodimerization of FHL and FHY1, resulting in Cre-mediated excision of DNA sequences at *loxP*-flanked sites and activation of target gene expression. **b** DNA recombination induced by a ΔFHL (1–116 aa)- and ΔFHY1 (1–116 aa)-mediated split-Cre recombinase system. **c** Schematic of the genetic configuration for split-Cre recombinase system. **d** Comparison among ΔFHL/ΔFHY1-, ΔFHY1/ΔFHL-, and Coh2/DocS-mediated split-Cre recombinase systems. Data in **b**, **d** are presented as means ± s.d.; *n* = 3 independent experiments, with three technical replicates for each. Statistical comparisons were performed by one-way ANOVA; NS, not significant, ****$p < 0.0001$. **e**–**g** DNA recombination induced by the split-Cre recombinase systems in wild-type C57BL/6 mice. **e** Schematic of the time-line for ΔFHL/ΔFHY1- and Coh2/DocS-mediated DNA recombination in mice. **f** Detailed images of bioluminescence in mice. **g** Quantification of bioluminescence signals in **f**. **h**–**m** DNA recombination induced by the split-Cre recombinase systems in transgenic Cre-tdTomato reporter mice. **h** Schematic of the timeline for ΔFHL/ΔFHY1- and Coh2/DocS-mediated DNA recombination activity in mice. **i** Detailed images of the tdTomato fluorescence in isolated liver tissue. **j** Quantification of the fluorescence signals in **i**. **k**, **l** qPCR (**k**) and western blot (**l**) analysis of tdTomato signal in isolated liver tissues from (**i**). **m** Representative fluorescence images of the liver sections of the transgenic Cre-tdTomato reporter mice shown in **i**. The images represent typical results from three independent measurements. Scale bar, 50 μm. Data in **g**, **j** are presented as means ± s.e.m.; *n* = 4 mice, in **k** are presented as means ± s.e.m.; *n* = 6 mice. Statistical comparisons were performed by one-way ANOVA; NS, not significant, *$p < 0.05$, **$p < 0.01$, ***$p < 0.001$, ****$p < 0.0001$. Source data are provided as a Source Data file.

We next assessed the dynamics of target gene expression in HEK293T cells expressing the PTRC transcriptional regulation system. Over 48 hours, FHL- and FHY1-induced SEAP production was consistently inhibited under constant illumination with red light (660 nm, 1 mW/cm²) (Fig. 3d and Supplementary Fig. 9b). Moreover, the extent of transcriptional repression depended on the duration and intensity of red-light exposure (Fig. 3e, f). To investigate the long-term dynamics of light-responsive transcriptional regulation, we delivered the PTRC transcriptional regulation system into HEK293T cells using INVI DNA Transfection Reagent to enhance plasmid stability[39]. Continuous exposure to red light (660 nm, 1 mW/cm²) for more than 6 days effectively suppressed FHL-induced SEAP production (Supplementary Fig. 10), illustrating the system's capability for sustained transcriptional control. Moreover, the PTRC system demonstrated fully reversible kinetics over several cycles (Fig. 3g), highlighting its robustness and versatility for dynamic and precise gene expression control across various applications.

To determine the time needed to fully reverse transcriptional activation in transfected HEK293T cells, we first induced activation by keeping the cells in darkness for 12 hours. Subsequently, the cells were either exposed to red light (660 nm, 1 mW/cm²) or kept in continuous darkness. Hourly measurements of *SEAP* mRNA levels showed that transcriptional activation was immediately suppressed upon exposure to red light, with transcriptional deactivation stabilizing within about 5 hours (Fig. 3h). These results indicated that PTRC-mediated transcriptional repression could be switched off or on by alternating exposure to red or far-red light, suggesting potentially tunable, temporal regulation of gene expression.

To evaluate whether this PTRC-based system could also exhibit transcriptional repression function in vivo, we transfected plasmids encoding ΔPhyA-2×ZIM3, Gal4-FHL, and the corresponding luciferase reporter into mouse liver by hydrodynamic injection of the tail vein. Control mice were injected with pcDNA3.1(+) instead of Gal4-FHL. The mice were then intraperitoneally injected with PCB and exposed to red light (660 nm, 5 mW/cm², 1 minute on, 5 minutes off, alternating) or continuous dark (Fig. 3i). Measurement of luciferase expression by IVIS revealed that mice harboring the PTRC system exhibited significantly stronger bioluminescence signal compared to control mice that did not express Gal4-FHL (Fig. 3j and Supplementary Fig. 11). Further, red-light illumination resulted in a >40-fold decrease in bioluminescence signal, indicating that the Luciferase gene was effectively repressed in these mice (Fig. 3j and Supplementary Fig. 11). These cumulative results indicated that the PTRC transcriptional regulation system could enable both tunable and reversible control of target gene expression in mammalian cells, both in vitro and in vivo.

A PTRC-controlled CRISPRa (PTRC$_{dcas}$) system for regulating endogenous gene transcription.

CRISPR-mediated transcriptional activation (CRISPRa)-based regulation of gene expression has been used in various research and biomedical applications, including cell state reprogramming[4,40,41], cancer modeling[42], and therapeutics for genetic diseases[43]. Based on the high demand and broad applicability of CRISPRa, we hypothesized that PTRC could be used to improve its controllability for specific uses in transcriptional regulation research. We constructed a PTRC-mediated CRISPRa system (PTRC$_{dcas}$) by introducing FHL/FHY1 into the synergistic activation mediator (SAM) system[7], which reportedly enhances the efficiency of gene activation using a single guide RNA (sgRNA) bearing MS2 RNA aptamers. In dark conditions, the dCas9-sgRNA complex targets the gene of interest, leading to sTF MCP-FHL binding to the MS2 RNA aptamers and subsequent activation of endogenous target gene transcription. Upon exposure to red light (660 nm), the transcriptional repressor (ΔPhyA-2×ZIM3) specifically binds to MCP-FHL, blocking the activation of the endogenous gene (Fig. 4a).

We selected control of endogenous rhox homeobox family member 2 (*RHOXF2*) activation as a proof-of-concept test for the PTRC$_{dcas}$ system (Fig. 4b). To improve the robustness of the system, we also incorporated the MCP-FHL with an N-terminal intrinsically disordered region (IDR) from the human oncogene, FUS (FUSn; amino acids 1 to 214), to drive the liquid-liquid phase separation (LLPS) of the sTF and subsequent formation of TF droplets around the target promoters[44,45]. As expected, the FHL-FUS fusion protein triggered a substantial increase in transcription (Fig. 4b). To test the generalizability of the PTRC$_{dcas}$ system, we deployed MCP-FUS-FHL to activate the transcription of other, unrelated, endogenous genes, including the achaete-scute homolog 1 (*ASCL1*), myocardial infarction-associated transcript (*MIAT*), and titin (*TTN*) (Supplementary Fig. 12). Moreover, MCP-FUS-FHL can also be captured by ΔPhyA under red light illumination to terminate the endogenous gene transcription (Fig. 4c).

To test whether PTRC$_{dcas}$ could control endogenous gene transcription in vivo, mice were hydrodynamically injected with PTRC$_{dcas}$-encoding plasmids via the tail vein. At eight hours after injection, the mice were illuminated with red light (660 nm, 5 mW/cm², 1 minute on, 5 minutes off, alternating) or maintained in the dark, and qPCR was used quantify *Hbb* (hemoglobin subunit beta), *Ins2* (insulin 2), *Dkk1* (dickkopf WNT signaling pathway inhibitor 1), and *Ascl1* (achaete-scute family BHLH transcription factor 1) expression in liver tissues at 24 hours after hydrodynamic injection (Fig. 4d). As expected, MCP-FUS-FHL expression resulted in significant transcriptional upregulation of all four endogenous genes (Fig. 4e, f), which could be terminated by exposure to red light (Fig. 4f). Collectively, these results demonstrated that the PTRC$_{dcas}$ system could be used to precisely control the transcription of user-selected endogenous genes of interest in mammalian cells and in mice in vivo.

**Small molecule- and red light-induced transcriptional regulation in mice**

In light of the above improvements to a gene editing system, we examined whether PTRCs could enhance the controllability of

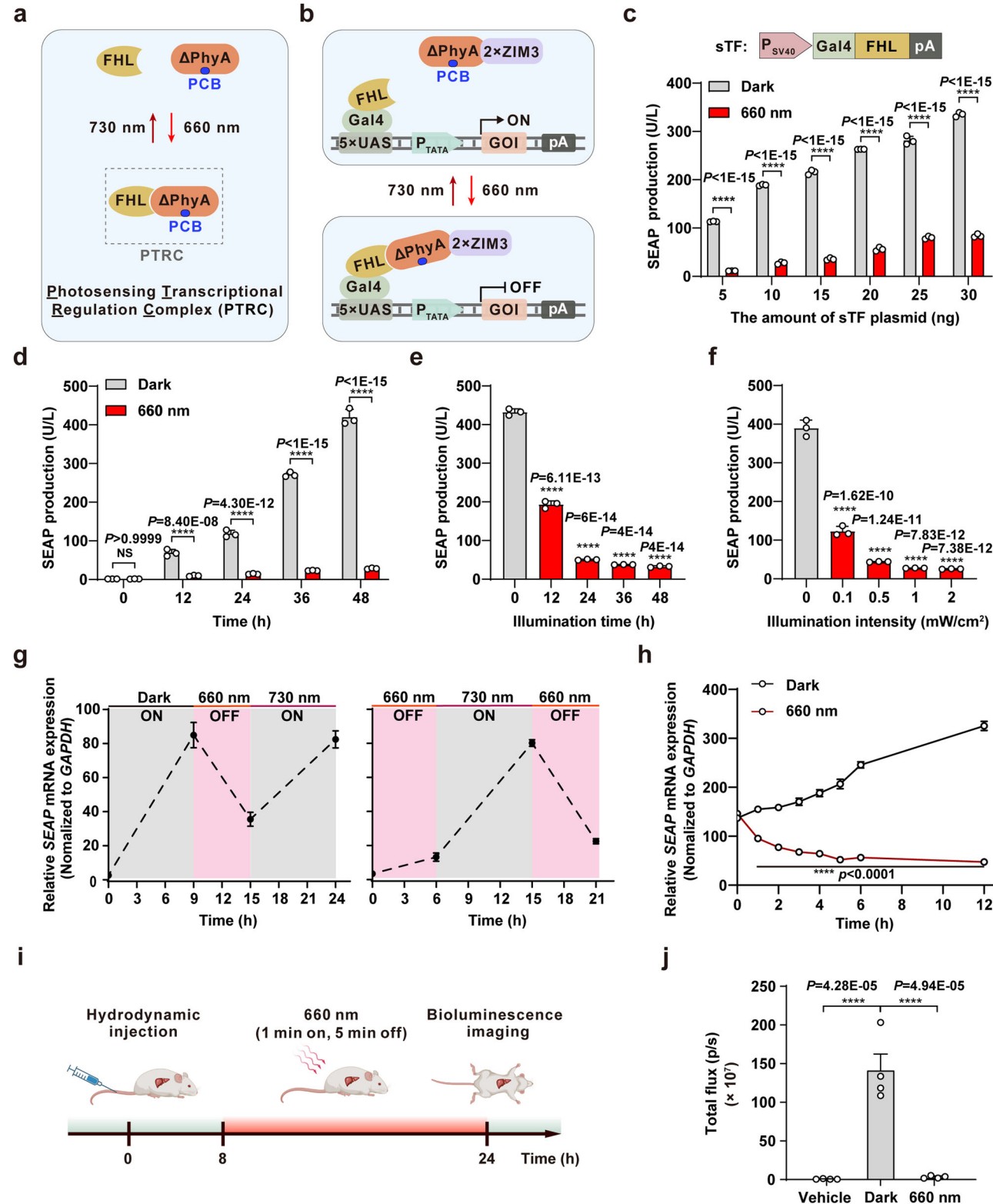

chemically induced proximity (CIP)-based transcriptional regulation systems. For this purpose, we propose a generally applicable strategy based on integrating the PTRC platform. By replacing the original transactivator with FHL, we developed a PTRC$_{CIP}$ system in which the small molecule-induced activation of a target gene could be promptly turned off under red light illumination.

To demonstrate the adaptability of FHL in various CIP-based transcriptional regulation systems, we constructed multiple PTRC$_{CIP}$ systems, including PTRC$_{CIP-ABA}$ (Abscisic acid-based PTRC$_{CIP}$ system), PTRC$_{CIP-DNV}$ (Danoprevir-based PTRC$_{CIP}$ system), PTRC$_{CIP-GZV}$ (Grazoprevir-based PTRC$_{CIP}$ system), and PTRC$_{CIP-RAPA}$ (Rapamycin-based PTRC$_{CIP}$ system), which consist of different synthetic transactivators, such as PYL1-FHL, NS3a-FHL, and FRB-FHL[46]. In the presence of small molecules (ABA, DNV, GZV, or RAPA), the synthetic DBD binds to its corresponding promoter and recruits the specific small molecule-responsive transactivator to initiate target gene expression

**Fig. 3 | Construction and characterization of a PTRC-mediated transcriptional regulation system in mammalian cells and mice. a** Schematic of a photosensing transcriptional regulation complex (PTRC). **b** Schematic design of a PTRC-mediated transcriptional regulation system. In dark conditions, sTF (Gal4-FHL) binds to its synthetic promoter (5×UAS-P$_{TATA}$) to initiate the expression of the gene of interest (GOI). Upon exposure to red light (660 nm), ΔPhyA specifically binds to FHL in the presence of the chromophore PCB, forming a PTRC and consequently suppressing FHL transcriptional regulatory activity. Subsequent illumination with far-red light (730 nm) induces ΔPhyA dissociation from FHL, resulting in PTRC disassembly and restoration of target gene expression. **c** Quantification of PTRC-mediated gene expression. HEK293T cells transfected with light-responsive transcriptional repressor (ΔPhyA-2×ZIM3), SEAP reporter (5×UAS-P$_{TATA}$-SEAP-pA), and different amounts of plasmids encoding Gal-FHL were illuminated with red light (660 nm, 1 mW/cm$^2$) for 24 hours, and then SEAP production was quantified. **d** Quantification of PTRC-mediated gene expression dynamics. **e** Exposure time-dependent gene expression dynamics. **f** Illumination intensity-dependent gene

expression dynamics. **g** Reversibility of PTRC-mediated transcriptional activation. Gray shading represents transcriptional activation (ON), while red shading represents transcriptional deactivation (OFF). **h** Dynamics of red light-induced transcriptional deactivation. Transfected cells were first maintained in darkness for 12 hours and then either exposed to red light (660 nm, 1 mW/cm$^2$) or kept in continuous darkness. The relative mRNA expression levels of *SEAP* were quantified using qPCR at intervals of 0, 1, 2, 3, 4, 5, 6, and 12 hours following the onset of red-light exposure. Data in **c**–**h** are presented as means ± s.d.; *n* = 3 independent experiments, with 3 technical replicates for each. Statistical comparisons in (**c**, **d**, **h**) were performed by two-way ANOVA, in (**e**, **f**) were performed by one-way ANOVA; NS, not significant, ****$p < 0.0001$. **i** Schematic of the timeline for PTRC-mediated gene expression in mouse livers. **j** Quantification of the bioluminescence signals in mouse livers. Data in **j** are presented as means ± s.e.m.; *n* = 4 mice. Statistical comparisons were performed by one-way ANOVA; ****$p < 0.0001$. Source data are provided as a Source Data file.

(Supplementary Fig. 13a). Upon red light illumination, ΔPhyA-2×ZIM3 specifically binds to the synthetic transactivator and terminates target gene expression (Fig. 5a). Tests of SEAP production showed that the PTRC$_{CIP}$ system could indeed mediate small molecule-inducible target gene expression (Supplementary Fig. 13b–e), which also could be terminated by red light illumination (Fig. 5b–e).

Moreover, the red light-induced transcriptional deactivation mediated by the PTRC$_{CIP}$ system occurred significantly faster than natural dissociation processes, which are often mediated by small molecule degradation (Fig. 5f, g). This finding indicates the potential of the PTRC$_{CIP}$ system for in vivo applications, particularly in scenarios where the rapid removal of small molecules is not practical.

We then assessed PTRC$_{CIP}$ performance in live mice by delivering plasmids encoding the PTRC$_{CIP-ABA}$ system to the mouse liver via hydrodynamic tail vein injection. The mice were intraperitoneally injected with ABA eight hours later and then illuminated with red light or kept in darkness. The bioluminescence signal was measured using an IVIS at 24 hours after hydrodynamic injection (Fig. 5g), which revealed that ABA-treated mice exhibited significantly higher luciferase expression than untreated controls and that the signal dramatically decreased in mice upon exposure to red light (Fig. 5h and Supplementary Fig. 14). Collectively, these results showed that PTRC$_{CIP}$ systems can be used to efficiently regulate target gene expression in mammalian cells and in mice triggered by small molecules, which can be rapidly terminated through red light illumination.

## Dual-wavelength light-controlled transcriptional regulation in mice

Blue light-activatable optogenetic switches such as CRY2-CIBN (cryptochrome 2-based)[47], iLID (AsLOV2-based)[48], and Magnets systems (pMag- and nMag-based)[49] have been widely employed to manipulate gene and protein activity in mammalian cells[50]. They offer non-invasiveness and dose-dependence control, making them valuable for various applications in fundamental and biomedical research[45]. To test whether FHL could be used with existing optogenetic systems to control transcriptional activation and deactivation, we combined a blue light responsive cryptochrome (CRY2-CIBN) system[47] with the PTRC, resulting in a dual-wavelength light-controlled transcriptional regulation system, PTRC$_{DL}$. For this purpose, we fused FHL with the PHR domain of CRY2 (CRY2$_{PHR}$) to create a synthetic dual-wavelength light-inducible transactivator (CRY2$_{PHR}$-FHL). Under blue light illumination, CRY2$_{PHR}$-FHL could be recruited by a synthetic DNA-binding domain (CIBN-Gal4) to a chimeric promoter (5×UAS-P$_{TATA}$) to initiate target gene expression. Importantly, target gene expression could be quickly deactivated by red light (Fig. 6a). Our results verified that PTRC was compatible with the CRY2-CIBN system, indicated by a significant increase in SEAP production (Fig. 6b). To verify that PTRC$_{DL}$-mediated target gene expression could be quickly deactivated by red light

illumination, we divided transfected HEK293T cells into five groups, including Group A, which was kept in darkness; Group B was illuminated with red light (660 nm, 1 mW/cm$^2$) for 48 h; Groups C, D, and E was illuminated with blue light (465 nm, 1 mW/cm$^2$) for 12 hours to initiate target gene expression, followed by blue light (Group C), stored in the dark (Group D), or illumination with red light (660 nm, 1 mW/cm$^2$) for 36 hours (Group E) (Fig. 6c). Luciferase reporter expression revealed that red light illumination could effectively and rapidly switch off blue light-induced target gene expression, and that expression levels significantly decreased, reaching similar levels to background (Group A and B), indicating complete deactivation of target gene expression (Fig. 6d). Compared to the natural dissociation processes mediated by dark reversion, the red light-induced transcriptional deactivation by the PTRC$_{DL}$ system occurred significantly faster (Fig. 6e, f). This highlights the efficiency and potential of the PTRC$_{DL}$ system for rapid control of gene expression.

To evaluate the potential of the PTRC$_{DL}$ system for in vivo application, plasmids encoding the PTRC$_{DL}$ system were delivered into the mouse liver through hydrodynamic tail vein injection. Eight hours after injection, mice were again divided into four groups. Control mice were kept in the dark for 16 hours. Mice in groups A, B, and C were illuminated with blue light (465 nm, 5 mW/cm$^2$) for 4 hours. In addition, group A was continuously illuminated with blue light for 12 hours, group B was stored in the dark, while group C was continuously illuminated with red light (660 nm, 1 mW/cm$^2$) for 12 hours (Fig. 6g). Measurement of Luciferase expression at 24 hours after injection by IVIS indicated that PTRC$_{DL}$-mediated target gene expression could be effectively turned on or off by blue or red light, respectively, in vivo (Fig. 6h, i). These findings highlighted the compatibility of PTRC in accommodating different inducible systems, and its potential for enhancing the tunability of these systems through simple substitutions of the transactivator component.

## Discussion

Here, we provide, to our knowledge, the first example with proof-of-concept application in which plant-derived chaperone proteins, FHL/FHY1, were introduced as transactivators in sTFs to control target gene expression in mammalian cells. Through DNA truncation, we identified the DNA regions required for transcriptional activation and homo-dimerization or heterodimerization by FHL/FHY1. It is worth noting that the FHL/FHY1 activation region overlaps with the region responsible for PhyA binding. Based on these findings, we constructed a versatile light-switchable platform for controlling transcriptional activation and deactivation. This platform is highly modular and compatible with other molecular tools, and we first implemented it through a PTRC-mediated gene expression system and a PTRC-mediated CRISPR-dCas9 system, which enabled transcriptional activation and red light-inducible deactivation of both exogenous and endogenous

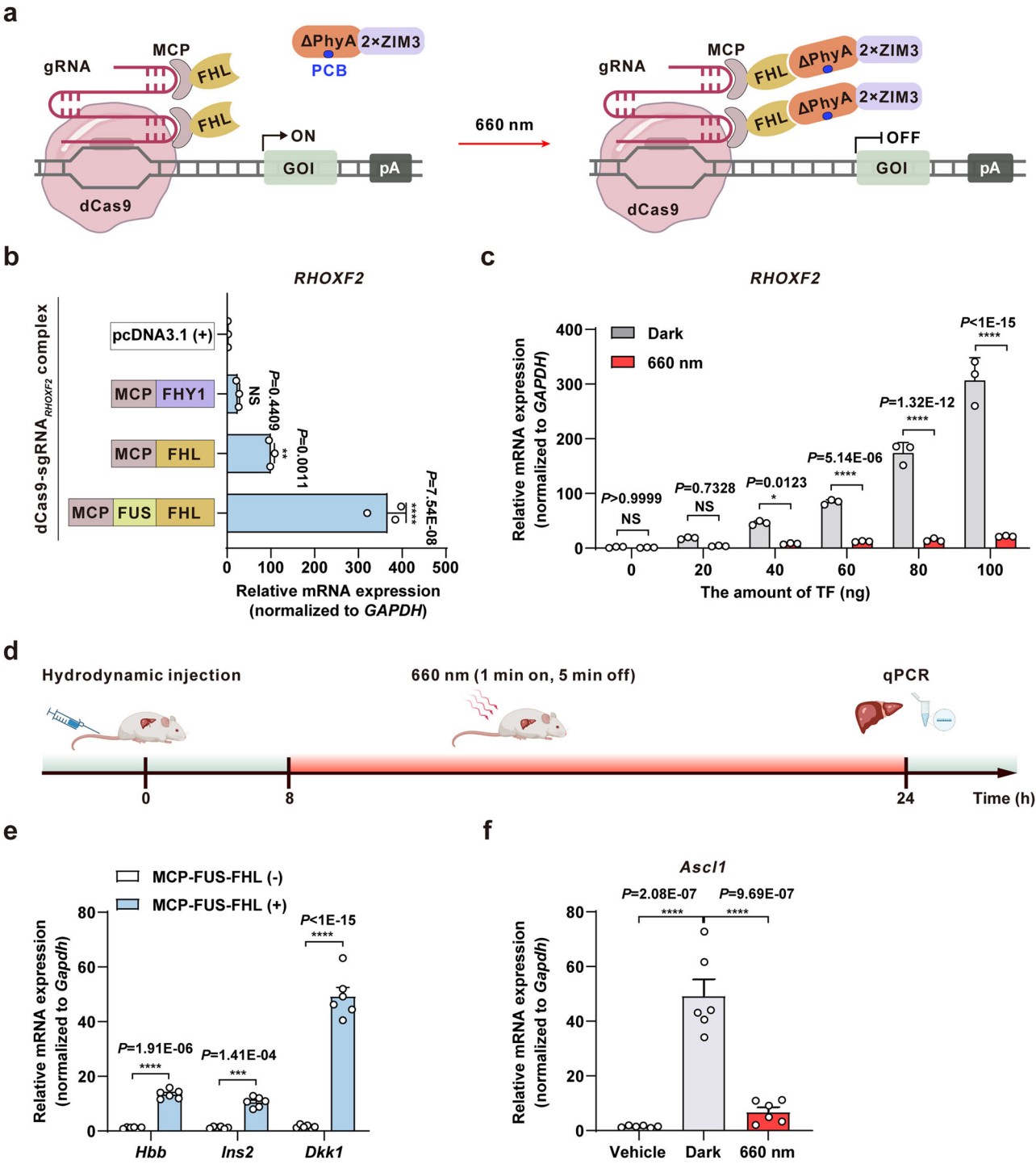

genes in multiple mammalian cells and mice. We then further expanded the utility of the PTRC by incorporating it into existing small molecule- or blue light-inducible transcriptional regulation systems. By simply replacing the original transactivator with FHL/FHY1, we have successfully developed a modified transcriptional regulation system that can be turned off through exposure to red light, resulting in a highly adaptable, flexible, and reversible system for controlling gene expression.

Compared to currently available transactivators, FHL/FHY1 can provide greater tunability, especially in vivo, where prompt gene transcription termination is often challenging. More specifically, other light- or small molecule-inducible systems cannot be readily inactivated due to the difficulty of removing small molecules or the

relatively slow dissociation kinetics of photosensitive proteins[18,47]. The introduction of PTRC thus enhances the tunability of transcriptional regulation systems and provides a user-defined means of deactivation through red light illumination.

Although we explored only a limited number of combinations thus far, the PTRC-based platform for transcriptional regulation shows high modularity, allowing the possible replacement of primary regulation systems and DBDs in FHL/FHY1-based sTFs. This flexibility eliminates the need for extensive system engineering and customization for each specific application, providing a straightforward approach to increasing controllability. Thus, various transcriptional regulation systems can be programmed with dual control mechanisms for use in mammalian cells, offering enhanced versatility and

**Fig. 4 | A PTRC-controlled CRISPR-dCas9 (PTRC$_{dcas}$) system for endogenous gene transcriptional regulation in mammalian cells and mice. a** Schematic of a PTRC-controlled CRISPR-dCas9 (PTRC$_{dcas}$) system. In dark conditions, the dCas9-sgRNA complex targets the gene of interest (GOI), leading to sTF (MCP-FHL) binding to the MS2 RNA aptamers and subsequent activation of endogenous target gene transcription. Upon exposure to red light (660 nm), the transcriptional repressor (ΔPhyA-2×ZIM3) specifically binds to MCP-FHL, blocking the activation of the endogenous gene. **b** Quantification of endogenous gene activation mediated by different sTF. HEK293T cells transfected with plasmids encoding sgRNA$_{RHOXF2}$-dCas9 complex and different sTFs were cultivated for 48 hours, and the relative mRNA expression of *RHOXF2* was quantified by qPCR. **c** Quantification of endogenous gene activation and deactivation mediated by PTRC$_{dcas}$ system with different amounts of MCP-FUS-FHL. HEK293T cells transfected with ΔPhyA-2×ZIM3, sgRNA$_{RHOXF2}$-dCas9 complex, and different amounts of MCP-FUS-FHL were illuminated with red light (660 nm, 1 mW/cm$^2$) for 48 hours or kept in darkness, and

then the relative mRNA expression of *RHOXF2* was quantified by qPCR. Data in **b**, **c** are presented as means ± s.d.; $n = 3$ independent experiments, with 3 technical replicates for each. Statistical comparisons in **b** were performed by one-way ANOVA, in **c** were performed by two-way ANOVA; NS, not significant, *$p < 0.05$, **$p < 0.01$, ****$p < 0.0001$. **d** Schematic of the timeline for PTRC$_{dcas}$-mediated endogenous gene activation and deactivation in mice. **e** MCP-FUS-FHL-mediated multiple endogenous gene activation in mice. Plasmids encoding dCas9, the MCP-FUS-FHL, and sgRNAs targeting different endogenous genes (*Hbb*, *Ins2*, and *Dkk1*) were delivered into the mouse livers by hydrodynamic injection via the tail vein. Relative mRNA expression was quantified by qPCR 24 hours after injection. **f** qPCR analysis of PTRC$_{dcas}$-mediated *Ascl1* transcription in mouse livers as described in **d**. Negative control mice (Vehicle) were hydrodynamically injected with NTsgRNA. Data in **e**, **f** are presented as means ± s.e.m.; $n = 6$ mice. Statistical comparisons in **e** were performed by two-way ANOVA, in **f** were performed by one-way ANOVA; ***$p < 0.001$, ****$p < 0.0001$. Source data are provided as a Source Data file.

adaptability. The multifunctionality of FHL and FHY1 also opens up possibilities for developing more intricate and advanced multi-input logic operation tools, thus simplifying their construction.

We also validated the homodimerization and heterodimerization capabilities of FHL/FHY1 in mammalian cells. Building on these findings, we developed the FHL/FHY1-mediated split-Cre recombinase system. The truncated FHL/FHY1-mediated split-Cre recombinase demonstrated higher catalytic activity than the DocS/Coh2-mediated split-Cre recombinase, highlighting its significant potential for advancing the fields of DNA recombination research and genome engineering. Additionally, FHL/FHY1 is also believed to mediate the recombination of other split proteins, such as Flp recombinase[51], and TEV protease[52]. The strong binding affinity in FHL/FHY1 interactions motivated our further exploration of their use in more complex protein-protein interactions, including designing interactions responsive to post-translational modifications, as well as small molecule, peptide, and protein binding, which will be explored in future studies.

We recognize that extensive application of the PTRC requires some further testing and possible modification to adapt to different contexts. For instance, it may be necessary to further enhance the transactivation by FHL/FHY1 in therapies targeting clinical genes in vivo. Here, we discovered that introducing mutations to the SUMOylation sites of FHY1 could significantly enhance its transactivation function (Supplementary Fig. 15). Since we identified the key functional domains of FHL and FHY1 involved in its regulatory activity, more synthetic transactivators can be constructed by increasing the copy number of activation functional domains. Moreover, by integrating FHL/FHY1 with other transactivators or by increasing their concentration or residence time around target promoters via phase separation[45], we can further enhance the capacity for transcriptional regulation by FHL and FHY1.

Overall, this versatile, light-switchable, transcriptional regulation platform provides a powerful tool for orthogonal, modular, and tunable transcriptional regulation in mammals. We anticipate that robust transactivators and more transcriptional regulation platforms will be useful for various applications across life science research, biotechnology, and biomedical engineering applications.

## Methods
### Plasmid construction
Comprehensive design and construction details for all expression vectors are provided in Supplementary Data 1. The DNA sequences for PTRC components are provided in Supplementary Table 1. Some plasmids were constructed using a MultiS One Step Cloning Kit (C113-01, Vazyme) according to the manufacturer's instructions. All relevant genetic components have been confirmed by sequencing (Personalbio).

### Cell culture and transfection
Human embryonic kidney cells (HEK293T cells, CRL-1573, ATCC), baby hamster kidney cells (BHK-21; CCL-10, ATCC), telomerase-immortalized human mesenchymal stem cells (hMSC-TERT; SCRC-4000, ATCC), B16-F10 melanoma cells (CRL-6475, ATCC), HEK293-derived Hana3A cells engineered for constitutive expression of RTP1, RTP2, REEP1 and Gαολφ[53], human cervical adenocarcinoma cells (HeLa; CCL-2, ATCC), human hepatocellular carcinoma cells (Huh7; TCHu182, Chinese Academy of Sciences), human Retinal Pigment Epithelial cell (ARPE-19; Chinese Academy of Sciences), and the mouse fibroblast NIH-3T3 cell line (CRL-1658, ATCC) were cultured in Dulbecco's modified Eagle's medium (DMEM; C11995500BT, Gibco) supplemented with 10% (vol/vol) fetal bovine serum (FBS; 16000-044, Gibco) and 1% (vol/vol) penicillin/streptomycin solution (ST488-1/ST488-2, Beyotime). All cell lines were cultured at 37 °C in a humidified atmosphere containing 5% CO$_2$ and were regularly tested for the absence of mycoplasma and bacterial contamination.

HEK293T, hMSC-TERT, and Hana3A cells were transfected with an optimized polyethyleneimine (PEI)-based protocol. Unless explicitly indicated, all the cell experiments were performed as follows. Briefly, the cells were plated $6 × 10^4$ cells/well in a 24-well plate and cultured for 18 hours before transfection. Subsequently, cells were incubated for 6 hours with 50 μL of PEI and DNA mixture at a mass ratio of 3:1 (PEI, molecular weight 40,000, stock solution 1 mg/mL in double distilled water; 24765, Polysciences). For the transfection of B16-F10, HeLa, Huh7, ARPE-19, and NIH-3T3 cells, the cells were plated at $5 × 10^4$ cells/well in a 24-well plate and cultivated for 12 hours at the time of transfection using Lipofectamine 8000 transfection reagent (C0533FT, Beyotime) according to manufacturer's protocol. For transfection, HEK293T cells were plated at $5 × 10^4$ cells/well in a 24-well plate and cultivated for 12 hours. At the time of transfection, INVI DNA Transfection Reagent (IV1214, Invigentech) was used according to the manufacturer's protocol. Detailed transfection mixtures are provided in Supplementary Data 2 and 3.

### Small molecule induction
Resveratrol (50 mM; R5010, Sigma-Aldrich) was purchased from Sigma-Aldrich, prepared in 50 mM stock solutions in 100% DMSO, stored at −20 °C, and diluted to different working concentrations using DMEM medium. Stocks (100×) of ABA (10 mM; 90769, Sigma-Aldrich) and RAPA (10 μM; A606203, Sangon Biotech) were prepared in 100% ethanol and stored at −20 °C. A 100× stock of GZV (1 mM; A173303, Adooq Bioscience), DNV (1 mM; RG7227, Adooq Bioscience), and PCB (500 μM; P14137, Frontier Scientific) were made in DMSO and stored at −20 °C. These molecules were added to cell cultures, so the final concentration was 1× at induction time. For mouse experiments, ABA (200 mg/kg) and PCB (5 mg/kg) were intraperitoneally injected after being dissolved into 100 μL PBS (A600100, Sangon Biotech).

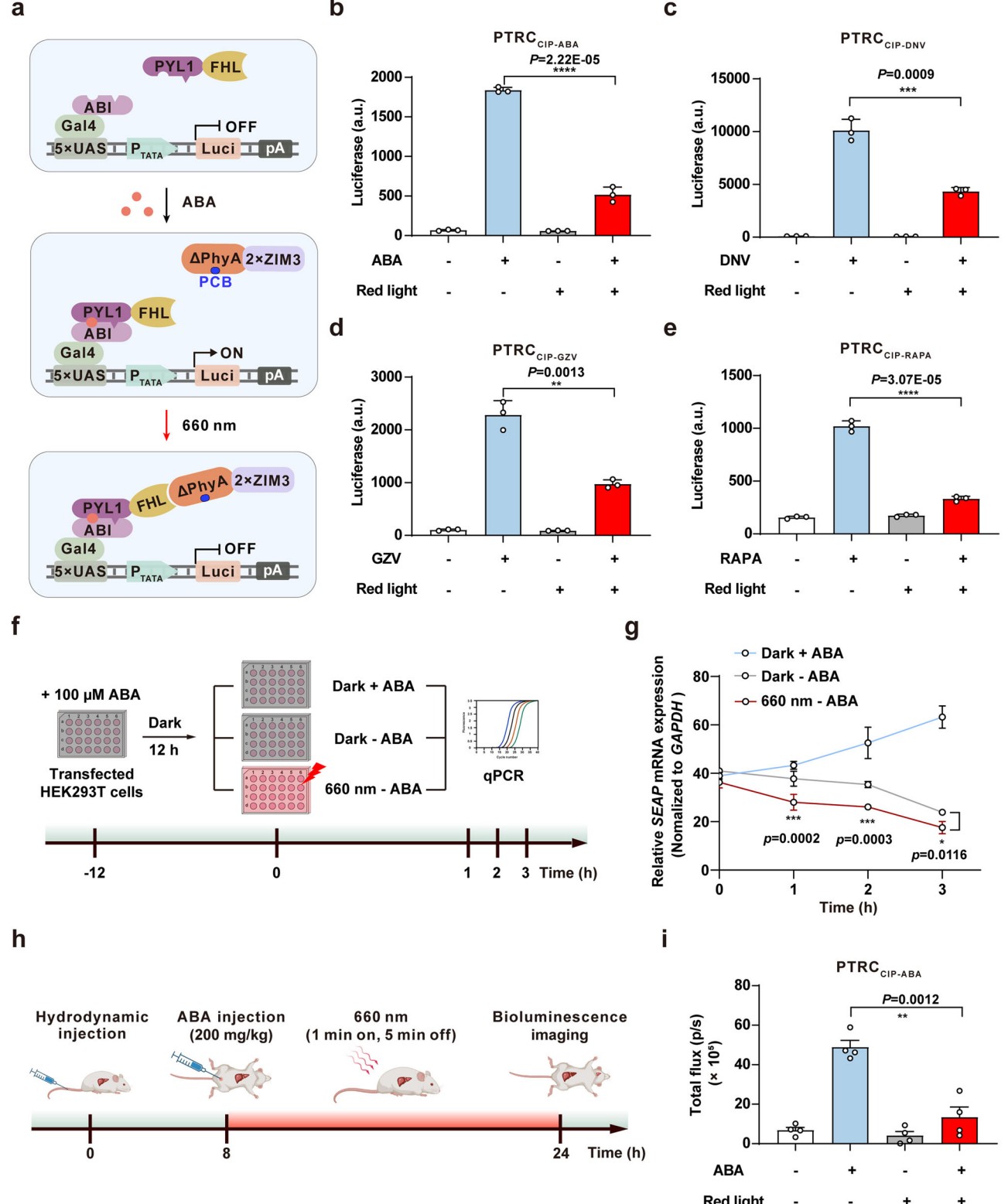

## Light illumination

For cell experiments, 24-well plates containing the samples were placed below a custom-designed LED array ($4 \times 6$) emitting blue, red, or far-red light (465 nm, 660 nm, or 730 nm; Shenzhen Bested Opto-electronic, with each LED centered above a single well). The illumination intensity was set to 1 mW/cm² at 660 nm, 1 mW/cm² at 730 nm, and 1 mW/cm² at 465 nm, unless explicitly indicated. This illumination process was carried out after transfection in 37 °C humidified

incubators. Plates not subjected to light treatment were wrapped in aluminum foil immediately after transfection. The light intensity was measured at a wavelength of 465 nm, 660 nm, or 730 nm using an optical power meter (Q8230, Advantest).

For mouse experiments, the devices and LED were sourced from Shenzhen Kiwi Lighting Co. Ltd. The LED beads had a power rating of 50 mW/cm², with available wavelengths of 465 nm or 660 nm. The light angle of the lamp was between 115 and 130°, which was narrowed to

Fig. 5 | A small molecule- and red light-induced transcriptional regulation (PTRC$_{CIP}$) system for controllable transcriptional activation and deactivation in mice. a Schematic of a small molecule- and red light-induced transcriptional regulation (PTRC$_{CIP}$) system. In the presence of abscisic acid (ABA), the synthetic DBD (ABI-Gal4) binds to its corresponding promoter (5×UAS-P$_{TATA}$) and recruits the ABA-responsive transactivator (PYL1-FHL) to initiate target gene expression. Upon exposure to red light (660 nm), ΔPhyA-2×ZIM3 specifically binds to FHL and terminates target gene expression. b–e The activation and deactivation performance of different PTRC$_{CIP}$ systems. HEK293T cells were transfected with abscisic acid-based PTRC$_{CIP}$ (PTRC$_{CIP-ABA}$: ABI-Gal4, PYL1-FHL, and ΔPhyA-2×ZIM3) (b), Danoprevir-based PTRC$_{CIP}$ (PTRC$_{CIP-DNV}$: DNCR2-Gal4, NS3a-FHL, and ΔPhyA-2×ZIM3) (c), Grazoprevir-based PTRC$_{CIP}$ (PTRC$_{CIP-GZV}$: Gal4-GNCR1, NS3a-FHL, and ΔPhyA-2×ZIM3) (d), or Rapamycin-based PTRC$_{CIP}$ (PTRC$_{CIP-RAPA}$: Gal4-FKBP, FRB-FHL, and ΔPhyA-2×ZIM3) (e). The cells were illuminated with red light (660 nm,

1 mW/cm$^2$) for 24 hours, and luciferase expression was quantified. f Schematic representation of the timeline for PTRC$_{CIP-ABA}$-mediated transcriptional activation. Gray shading represents placement in darkness, while red shading represents exposure to red light. g qPCR analysis of PTRC$_{CIP-ABA}$-mediated *SEAP* transcription as described in f. Data in b–e, g are presented as means ± s.d.; $n = 3$ independent experiments, with 3 technical replicates for each. Statistical comparisons in b–e were performed by two-tailed Student's *t* test, in g were performed by two-way ANOVA; NS, not significant, $*p < 0.05$, $**p < 0.01$, $***p < 0.001$, $****p < 0.0001$. *P* values in g were calculated by comparing the Dark–ABA group with the 660 nm–ABA group. h Schematic of the timeline for PTRC$_{CIP-ABA}$-mediated gene expression in mouse livers. i Quantification of bioluminescence signals in mice. Data in i are presented as means ± s.e.m.; $n = 4$ mice. Statistical comparisons were performed by two-tailed Student's *t* test; $**p < 0.01$. Source data are provided as a Source Data file.

60° upon the addition of a spotlight kit. The mice were exposed to illumination 8 hours post-injection, at an intensity of 5 mW/cm$^2$ (1 minute on, 5 minutes off, alternating) for 450 nm or at 10 mW/cm2 (1 minute on, 5 minutes off, alternating) for 660 nm, unless explicitly indicated.

## SEAP assay
The quantification of human placental SEAP in cell culture medium was conducted as previously reported[54]. Briefly, 120 μL of substrate solution [100 μL of 2×assay buffer containing 20 mM homoarginine (1483-01-8, Sangon Biotech), 1 mM MgCl$_2$, 21% (w/w) diethanolamine (pH 9.8), and 20 μL of substrate containing 120 mM p-nitro phenyl phosphate (333338-18-4, Sangon Biotech)] was added to 80 μL heat-inactivated (65 °C for 30 min) cell culture supernatant, and the light absorbance time course at 405 nm was measured using a Synergy H1 hybrid multimode microplate reader (BioTek Instruments) with Gen5 software (version 2.04). Unless explicitly indicated, SEAP production was detected 24 hours after transfection.

## Fluorescence imaging
Fluorescence imaging of EGFP expression in cells was performed with an inverted fluorescence microscope (Olympus IX71, TH4-200, Olympus) equipped with an Olympus digital camera (Olympus DP71, Olympus) and a 495/535-nm (blue/green/red) excitation/emission filter set, and images were acquired with 480 nm (excitation) and 535 nm (emission) filters and analyzed using Image-Pro Express C software (version ipp6.0) for EGFP signal.

## Luciferase reporter assay
Luciferase activity levels were measured using the Dual-Luciferase assay kit (RG005, Beyotime). Briefly, cell samples were treated with 200 μL cell lysis buffer per well of a 24-well plate for 5 minutes. Lysis supernatants were collected after centrifuged at 13,800 × g for 5 minutes. The mixture of 20 μL lysis supernatants and 20 μL luciferase substrate was added to the 96-well plate. The luminescence signal was detected using the Synergy H1 hybrid multi-mode microplate reader (BioTek Instruments).

## qPCR analysis
Cells were harvested for total RNA isolation using an RNAiso Plus kit (9109, Takara). A total of 1 μg RNA was reverse transcribed into cDNA using a PrimeScript RT Reagent Kit (RR047, Takara). qPCR reactions were performed on the LightCycler 96 real-time PCR instrument (Roche) using the SYBR Premix Ex Taq (RR420, Takara) for detecting each target gene, and the $2^{-\Delta\Delta Ct}$ method was used to calculate relative gene expression. The gRNA sequences used in this study are listed in Supplementary Table 2, and the qPCR primers used in this study are listed in Supplementary Table 3.

## Western blot analysis
Tissue samples were lysed in RIPA buffer (50 mM Tris-HCl, pH 7.5, 150 mM NaCl, 0.1% sodium deoxycholate, 0.1% SDS, 1 mM EDTA pH 8.0, 1% NP-40) containing 1 mM phenylmethanesulfonyl fluoride (PMSF). The mixture was ground into homogenate on ice, and the supernatant was collected by centrifuging at 16,200 × g for 15 minutes at 4 °C. The protein concentration in samples was determined using a bicinchoninic acid assay kit (P0012S, Beyotime). Lysates were mixed with loading buffer and boiled for 10 minutes. An equal amount of proteins (30 μg) were run on a 10% sodium dodecyl sulfate-polyacrylamide gel (SDS-PAGE) and then transferred onto a polyvinylidene fluoride (PVDF) membrane (IPVH00010, Millipore MA). The membrane was blocked with 5% nonfat milk in TBST buffer (50 mM Tris, 1.37 mM NaCl, 2.7 mM KCl, 0.05% Tween 20, pH 8.0) for 1 hour at room temperature. The membranes were then incubated with anti-tdTomato primary antibody (1:500; A00682, GenScript) or anti-GAPDH primary antibody (1:1000; AF1186, Beyotime) overnight at 4 °C. After washing three times with TBST buffer, the membrane was incubated with a secondary antibody (1:5000; Alexa Fluor790 Goat Anti-Rabbit, Jackson ImmunoResearch) for 1 hour at room temperature. After washing three times with TBST buffer, the membrane was visualized using a fluorescent Western blot imaging system (LI-COR Odyssey Clx).

## PTRC-mediated gene expression dynamics
To evaluate PTRC-mediated gene expression, HEK293T cells transfected with PTRC-mediated transcriptional regulation system [5 ng pDQ521 (P$_{SV40}$-Gal4-FHL-pA), 300 ng pDQ362 (P$_{hCMV}$-ΔPhyA-2×ZIM3-pA), and 100 ng pYZ430 (5×UAS-P$_{TATA}$-SEAP-pA)] were illuminated with red light (660 nm, 1 mW/cm$^2$) for different time periods (0-48 hours). SEAP production was quantified after illumination.

To evaluate exposure time-dependent transgene expression, HEK293T cells were transfected with a PTRC-mediated transcriptional regulation system. Twelve hours later, the transfected cells were illuminated with red light (660 nm, 1 mW/cm$^2$) for different time periods (0-48 hours). SEAP production was quantified at 60 hours after transfection.

To evaluate illumination intensity-dependent gene expression, HEK293T cells transfected with a PTRC-mediated transcriptional regulation system were illuminated with red light (660 nm) at different light intensities (0–2 mW/cm$^2$) for 48 hours, and then SEAP production was quantified.

To evaluate the long-term dynamics of light-responsive transcriptional regulation, the PTRC transcriptional regulation system was introduced into HEK293T cells using the INVI DNA Transfection Reagent. After transfection, the cells were either kept in darkness or exposed to red light (660 nm, 1 mW/cm$^2$). The production of SEAP was quantified daily before the culture medium was replaced with fresh medium.

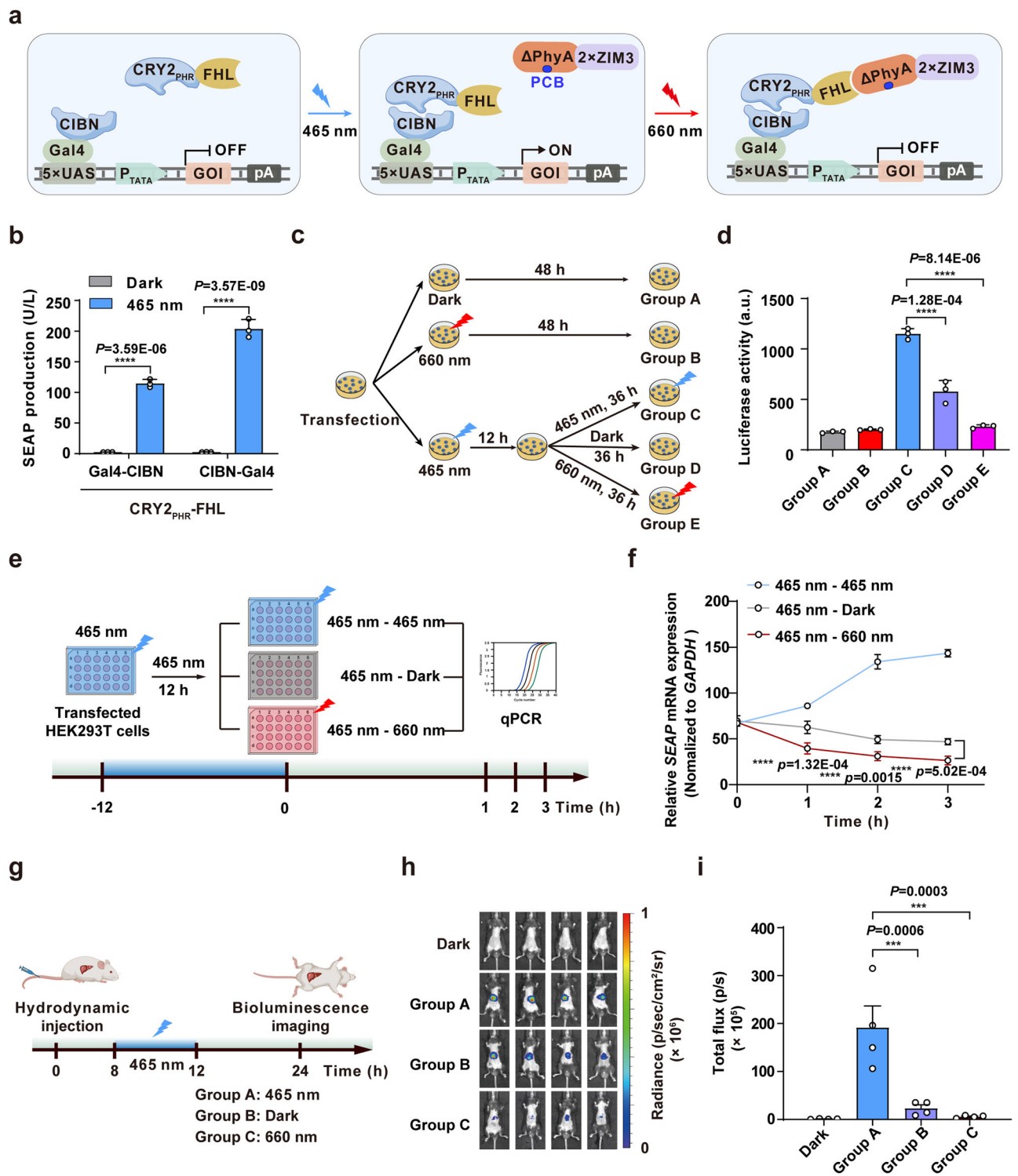

To evaluate the reversibility of the PTRC-mediated transcriptional regulation system, HEK293T cells transfected with this system were divided into two groups six hours after transfection. Cells in the ON-OFF-ON group were maintained in darkness for nine hours to activate transcription, followed by six hours of exposure to red light (660 nm, 1 mW/cm²) to deactivate it. Subsequently, these cells were exposed to far-red light (730 nm, 1 mW/cm²) for nine hours to reactivate transcription. Conversely, cells in the OFF-ON-OFF group were initially exposed to red light (660 nm, 1 mW/cm²) for six hours to deactivate transcription, then maintained in darkness for nine hours to activate it, and finally exposed again to red light (660 nm, 1 mW/cm²) for six hours

to deactivate it. The relative mRNA levels of SEAP were quantified by qPCR following changes in the treatment conditions.

To determine the time required to completely remove the transcriptional activation effect, HEK293T cells transfected with the PTRC-mediated transcriptional regulation system were first maintained in darkness for 12 hours to induce transcriptional activation. Subsequently, they were divided into two groups. Cells in the Dark group continued to be kept in darkness, whereas those in the 660 nm group were exposed to red light (660 nm, 1 mW/cm²). The relative mRNA levels of SEAP were quantified using qPCR after exposure to red light for intervals of 0, 1, 2, 3, 4, 5, 6, and 12 hours.

**Fig. 6 | A dual-wavelength light-controlled transcriptional regulation (PTRC$_{DL}$) system for controllable transcriptional activation and deactivation in mice.**
**a** Schematic of a dual-wavelength light-controlled transcriptional regulation (PTRC$_{DL}$) system. Under blue light (465 nm) illumination, the synthetic DBD (CIBN-Gal4) binds to its corresponding promoter (5×UAS-P$_{TATA}$) and recruits the dual-wavelength, light-responsive transactivator (CRY2$_{PHR}$-FHL) to initiate the target gene expression. Upon exposure to red light (660 nm), ΔPhyA-2×ZIM3 specifically binds to FHL and terminates target gene expression. **b** Quantification of the blue light-controlled transcriptional activation. HEK293T cells transfected with CIBN-Gal4 (or Gal4-CIBN), CRY2$_{PHR}$-FHL, and the SEAP reporter were illuminated with blue light (465 nm, 1 mW/cm$^2$) for 48 hours, and SEAP production was quantified. **c** Schematic of the timeline and experimental procedure for activation/deactivation performance of the PTRC$_{DL}$ system. **d** The activation and deactivation performance of the PTRC$_{DL}$ system in c. **e** Schematic representation of the timeline for PTRC$_{DL}$-mediated gene expression. Blue shading represents exposure to blue light, gray shading represents placement in darkness, and red shading represents exposure to red light. **f** qPCR analysis of PTRC$_{DL}$-mediated *SEAP* transcription as described in **e**. Data in **b**, **d**, **f** are presented as means ± s.d.; $n = 3$ independent experiments, with three technical replicates for each. Statistical comparisons in **b**, **f** were performed by two-way ANOVA, in **d** were performed by one-way ANOVA; NS, not significant, ****$p < 0.0001$. $P$ values in **f** were calculated by comparing the 465 nm−Dark group with the 465 nm−660 nm group. **g** Schematic of the timeline for dual-wavelength light-controlled gene expression in mouse livers. **h** Detailed images of bioluminescence in mice are described in **g**. **i** Quantification of bioluminescence signals in **h**. Data in **i** are presented as means ± s.e.m.; $n = 4$ mice. Statistical comparisons in **i** were performed by one-way ANOVA; ***$p < 0.001$. Source data are provided as a Source Data file.

## Assessment of the activation/deactivation performance of the PTRCCIP-ABA system

HEK293T cells transfected with the PTRC$_{CIP-ABA}$ system [100 ng pWY49 (P$_{hCMV}$-ABI-Gal4-pA), 100 ng pWY51 (P$_{hCMV}$-PYL1-FHL-pA), 100 ng pYZ430, and 200 ng pDQ362] were treated with 100 μM ABA in darkness for 12 hours before being divided into three groups. The first group was treated again with 100 μM ABA and kept in darkness (Dark + ABA). The second group was maintained in darkness without any additional ABA treatment (Dark - ABA), and the third group was exposed to red light (660 nm, 1 mW/cm$^2$) without any further ABA treatment (660 nm - ABA). The relative mRNA levels of *SEAP* were quantified using qPCR at 0, 1, 2, and 3 hours.

## Assessment of the activation/deactivation performance of the PTRC$_{DL}$ system

HEK293T cells transfected with the PTRC$_{DL}$ system [100 ng pDQ369 (P$_{hCMV}$-CIBN-Gal4-pA), 100 ng pDQ522 (P$_{hCMV}$-CRY2$_{PHR}$-FHL-pA), 100 ng pYZ450, and 200 ng pDQ362] were exposed to blue light (465 nm, 1 mW/cm$^2$) for 12 hours before being divided into three groups. The first group continued to be exposed to blue light (465 nm −465 nm), the second group was kept in darkness (465 nm - Dark), and the third group was exposed to red light (465 nm−660 nm). The relative mRNA levels of *SEAP* were quantified using qPCR at intervals of 0, 1, 2, and 3 hours.

## Animals

The experimental animals, including six-week-old C57BL/6 male mice and the transgenetic Cre-tdTomato reporter male mice (*Gt (ROSA) 26Sor$^{tm14(CAG-tdTomato)Hze}$*) were obtained from the ECNU Laboratory Animal Centre. All the animals were kept on a standard alternating 12-hour light/12-hour darkness cycle and given a normal chow diet [6% fat and 18% protein (wt/wt)] and water.

## FHY1/FHL-mediated DNA recombination in mice

For six-week-old C57BL/6 mice, two milliliters (10% of the body weight in grams) of Ringer's solution (147 mM NaCl, 4 mM KCl, 1.13 mM CaCl$_2$) containing a total of 125 μg plasmids encoding DocS/Coh2-mediated split-Cre recombinase system [50 μg pDQ532 (P$_{hCMV}$-DocS-CreC-pA), 50 μg pDQ533 (P$_{hCMV}$-CreN-Coh2-pA), and 25 μg pXY185 (P$_{hCMV}$-*loxp*-STOP-*loxp*-Luciferase-pA)], or ΔFHY1/ΔFHL-mediated split-Cre recombinase system [50 μg pDQ683 (P$_{hCMV}$-ΔFHY1-CreC-pA), 50 μg pDQ661 (P$_{hCMV}$-CreN-ΔFHL-pA) and 25 μg pXY185] was hydrodynamically injected via tail vein injection. Negative control mice (Vehicle) were hydrodynamically injected with pXY185. At 24 hours after plasmid injection, luciferase reporter expression was measured using an IVIS Lumina II in vivo imaging system (IVIS, PerkinElmer). For 6-week-old transgenetic Cre-tdTomato reporter mice, two milliliters (10% of the body weight in grams) of Ringer's solution containing a total of 200 μg plasmids encoding DocS/Coh2-mediated split-Cre recombinase system (100 μg pDQ532, 100 μg pDQ533), or ΔFHY1/ΔFHL-mediated split-Cre recombinase system (100 μg pDQ683, 100 μg pDQ661) was hydrodynamically injected via tail vein injection. Negative control mice (Vehicle) were hydrodynamically injected with pcDNA3.1(+). At seven days after plasmid injection, the mice were sacrificed, and the livers were isolated for fluorescence imaging, qPCR, Western blot, and histological analysis. The tdTomato signal from the isolated liver was detected using an IVIS equipped with tdTomato filter sets. The collected fluorescence emission signals were stored in epifluorescence units (radiance efficiency), and the total flux was calculated for ROI. All images were analyzed using the Living Image® 4.3.1 software.

## PTRC-mediated transcriptional regulation in mice

Two milliliters (10% of the body weight in grams) of Ringer's solution containing 248 μg plasmids [8 μg pDQ328 (P$_{hCMV}$-Gal4-FHL-pA), 160 μg pDQ362 and 80 μg pYZ450 (5×UAS-P$_{TATA}$-luciferase-pA)] were hydrodynamically injected into mice (male, six weeks old) via tail vein injection. Eight hours after plasmid injection, the mice were intraperitoneally injected with 5 mg/kg PCB and then exposed to red light (660 nm LED, 5 mW/cm$^2$, 1 minute on, 5 minutes off, alternating) for 16 hours. Negative control mice (Vehicle) were hydrodynamically injected with pYZ450. At 24 hours after plasmid injection, luciferase reporter expression was measured using an IVIS.

## PTRC$_{dcas}$-mediated regulation of endogenous gene transcription in mice

Two milliliters (10% of the body weight in grams) of Ringer's solution containing a total of 300 μg plasmids [100 μg pYZ561 (P$_{U6}$-sgRNA1$_{AscI1}$::P$_{U6}$-sgRNA2$_{AscI1}$::P$_{hCMV}$-dCas9-pA), 150 μg pDQ362 and 50 μg pDQ455 (P$_{hCMV}$-MS2-FUS-FHL-pA)] was hydrodynamically injected into mice (male, six weeks old) via tail vein injection. At 8 hours after plasmid injection, the mice were intraperitoneally injected with 5 mg/kg PCB and then exposed to red light (660 nm LED, 5 mW/cm$^2$, 1 minute on, 5 minutes off, alternating) for 16 hours. Negative control mice (Vehicle) were hydrodynamically injected with NTsgRNA. At 16 hours following illumination, the mice were sacrificed, and the livers were collected. RNA was extracted for qPCR analysis. The gRNA sequences used in this study are listed in Supplementary Table 2, and the qPCR primers used in this study are listed in Supplementary Table 3.

## PTRC$_{CIP}$-mediated transcriptional regulation in mice

Two milliliters (10% of the body weight in grams) of Ringer's solution containing 320 μg plasmids [80 μg pWY49 (P$_{hCMV}$-ABI-Gal4-pA), 40 μg pWY51 (P$_{hCMV}$-PYL1-FHL-pA), 40 μg pYZ450 and 160 μg pDQ362] were hydrodynamically injected into mice (6 weeks old) via tail vein injection. Eight hours after plasmid injection, the mice were intraperitoneally injected with 5 mg/kg PCB and 200 mg/kg ABA[55] and then exposed to red light (660 nm LED, 5 mW/cm$^2$, 1 minute on, 5 minutes

off, alternating) for 16 hours. At 24 hours after plasmid injection, luciferase reporter expression was measured using an IVIS.

### PTRC$_{DL}$-mediated transcriptional regulation in mice

Two milliliters (10% of the body weight in grams) of Ringer's solution containing 320 μg plasmids [80 μg pDQ369, 40 μg pDQ522, 40 μg pYZ450, and 160 μg pDQ362] were hydrodynamically injected into mice (6 weeks old) via tail vein injection. Eight hours after plasmid injection, the mice were intraperitoneally injected with 5 mg/kg PCB and then exposed to red light (660 nm LED, 5 mW/cm$^2$, 1 minute on, 5 minutes off, alternating) or blue light (465 nm LED, 5 mW/cm$^2$, 1 minute on, 5 minutes off, alternating) as indicated. At 24 hours after plasmid injection, luciferase reporter expression was measured using an IVIS.

### IVIS imaging

For in vivo imaging, each mouse was intraperitoneally injected with luciferin substrate solution (150 mg/kg; luc001, Shanghai Sciencelight Biology Science & Technology) intraperitoneally. Five minutes after the injection, bioluminescence images of the mice were captured using an IVIS. The bioluminescence images were then analyzed using Living Image software (version 4.3.1).

### Liver histology imaging

Fresh livers were washed three times with cold PBS to remove impurities such as blood and then fixed in 4% (w/v) paraformaldehyde (PFA; 30525-89-4, Sangon Biotech) for 2 hours at 4 °C. Tissue blocks of ~1 cm$^3$ were cut and embedded in an optimum cutting temperature compound (OCT; 03803389, Leica). Five μm thick liver sections were prepared using Cryostat Microtome (CM1950, Leica) and rinsed with PBS. Finally, samples were counterstained with 4′,6-diamidino-2-phenylindole (DAPI; 28718-90-3, Sigma) for 10 minutes. Endogenous gene tdTomato expression was observed on an inverted fluorescence microscope (DMI8, Leica).

### Ethics

The experiments involving animals were approved by the East China Normal University (ECNU) Animal Care and Use Committee and in direct accordance with the Ministry of Science and Technology of the People's Republic of China on Animal Care guidelines. The protocol (protocol ID: m20220412, m20220506) was approved by the ECNU Animal Care and Use Committee. All animals were euthanized after the experiments were terminated.

### Statistical analysis

All in vitro data are expressed as the mean ± SD of three independent experiments ($n = 3$). For the animal experiments, each treatment group consisted of randomly selected mice ($n = 4$–6). The results are expressed as mean ± SEM. Statistical significance was analyzed by the Student's $t$ test. Neither animals nor samples were excluded from the study. Differences were considered statistically significant at $p < 0.05$ (*), very significant at $p < 0.01$ (**), and extremely significant at $p < 0.001$ (***). GraphPad Prism software version 6.0 was used for statistical analysis.

### Reporting summary

Further information on research design is available in the Nature Portfolio Reporting Summary linked to this article.

## Data availability

All data associated with this study are presented in the paper or the Supplementary Information. All genetic components related to this paper are available with a material transfer agreement and can be requested from H.Y. (hfye@bio.ecnu.edu.cn). Source data are provided with this paper.

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

## Acknowledgements

This work was financially supported by grants from the National Natural Science Foundation of China (NSFC: no. 32250010, no. 32261160373), the National Key R&D Program of China, Synthetic Biology Research (no. 2019YFA0904500), the Science and Technology Commission of Shanghai Municipality (no. 23HC1410100 and 22N31900300), the Fundamental Research Funds for the Central Universities, and the Open Research Project of Shanghai Key Laboratory of Diabetes Mellitus (SHKLD-KF-2201) to H.Y. This work was also partially supported by the National Natural Science Foundation of China (no. 32301217) to J.Y. and the Young Scientists Fund of the National Natural Science Foundation of China (no. 32300458), the Science and Technology Commission of Shanghai Municipality (no. 23YF1410700), and China Postdoctoral Science Foundation (no. 2022M721163 and no. BX20230128) to Y.Z. We also thank the support from the CAS Youth Interdisciplinary Team and the ECNU Multifunctional Platform for Innovation (011) for supporting the mice experiments and the Instruments Sharing Platform of the School of Life Sciences, ECNU.

## Author contributions

H.Y. conceived the project. H.Y., G.N., D.K., Y.Z., and Y.W. designed the experiments, analyzed the results, and wrote the manuscript. D.K., Y.Z., Y.W., X.W., Q.H., X.G., H.W., M.L., L.K., G.Y., and L.J. performed the experimental work. D.K., Y.Z., Y.W., and G.N. designed, analyzed, and interpreted the experiments. All authors edited and approved the manuscript.

## Competing interests

The authors declare no competing interests.
