## [Peer Review File · Nature Communications]

Reviewers' Comments:

Reviewer #1:

Remarks to the Author:

The author's manuscript presents an approach in synthetic biology for precise control of gene expression through synthetic transcription factors (sTFs). The authors validated the transactivator functions of two plant-derived phytochrome chaperone proteins (FHY1 and FHL) in mammalian cells and mice, forming a photosensing transcriptional regulation complex (PTRC). This complex can toggle between active and inactive states through exposure to red or far-red light. Leveraging this capability, the author constructed a light-switchable platform for orthogonal, modular, and tunable control of gene transcription, which successfully modulated both exogenous and endogenous genes in multiple mammalian cells and mice. This manuscript is original and provides a promising tool for synthetic biology and biomedical engineering. A few comments for this article are listed below:

- 1) Line 60-63: The description in this section is unclear and confusing in conveying the intended viewpoint. Please check it carefully and provide more comprehensive references to synthesize the current research gap.
- 2) Line 84-85: Please supplement this section with references or list relevant works.
- 3) Line 128-130: The statement mentioning that FHY1 and FHL exhibited greater activation of SEAP expression than VP16 contradicts the results presented in Figure 1b. Please check it for accuracy.
- 4) Please indicate in the legend of Figure 1h and 1i the specific regions removed in each truncation variant.
- 5) Line 401-403: please check carefully whether this is the first article published to use FHL/FHY1 as transactivators in mammalian cells.
- 6) What is the long-term behaviour of PTRC system? How long will it take to completely remove the transcriptional activation effect?
- 7). Results showed in figure 3 indicated that this PTRC transcriptional regulation system is reversible by altering the illumination conditions. Is this reusable for multiple cycles? Please provide more details and discuss it.

Reviewer #2:

Remarks to the Author:

In the presented manuscript the authors engineered synthetic transcription factors (sTFs) using the two plant-derived phytochrome chaperones FHY1 and FHL as transactivators. After investigating the homo- and heterodimerization capacity of the chaperones, DNA regions mandatory for proper protein functionality were identified. The authors then used the heterodimerization of the two chaperones to establish a split-Cre recombinase system, showing enhanced activity over the already known DocS/Coh2 approach. Afterwards, through reversible binding of PhyA and the sTF, formation of a photosensing transcriptional regulation complex was investigated and it was demonstrated, that the complex can be controlled by red, far-red light exposure. Finally, the authors presented integration of this system in different already established tools, like the CRISPRa system or small molecule- or blue light inducible modules. All in all, the presented approaches enabled reversible control of transcriptional regulation in vitro and in vivo. In general, the study is presented in a comprehensive way and the claims made are supported by the data presented. Sometimes the inconsistency when displaying the fold-change or the significance in the same graph made it hard to interpret the given results and compare the individual datasets.

As the manuscript presents tools on protein-protein interaction as well as transcriptional regulation it can be of interest to a broad community in the fields of protein biology and transcriptional regulation, working in vitro and in vivo.

Even though the general concepts of the methods presented in the study already exist, the

improved functionality and multiple combination possibilities shown for the tools generated, highlight their practicability in synthetic biology and biomedical engineering.

In order to better understand the manuscript and the presented data, it is suggested to address the following points

Major:

- As the speed of the system is mentioned several times as an advantage, it might be of interest to show some kinetic graphs, presenting and comparing the actual time-dependent activity. Optimally, a comparison is performed to previously published systems in order to evaluate the performance of the present approach compared to the state of the art.
- A key feature of light-controlled systems is the reversibility. It is recommended to experimentally demonstrate the kinetics of the reversibility of the system

Minor

- Easy comparison of conditions is not possible if either significance or fold-change is given for a data set in the same graph, stick to one or present both (eg. 3c (statement not supported by data?), 9a,b)
- Addition of the number of technical replicates performed to the figures
- Figure 1a and S1a are identical
- Figure 1 statistical test used not indicated
- Figure 2m not really visible, maybe enhance quality or zoom in
- In Figure S4b, the meaning of the red box is not indicated and from line 203 on, a $\Delta FHL/\Delta FHY1$ version is mentioned, that is not clearly defined anywhere (equivalent to red box/ $\Delta 5$?), also it might be nice to add S4b to the main Figures, to improve understanding
- Line 199 typo (split-Cre), also the sentence is not really clear to me
- Figure S8b, there is already a significant drop stated without any plasmid added? Supposed to be 10 ng instead of 0?
- Figure 6, title: PTRCDL instead of PSTDL?

Point-by-Point Response

Manuscript number: NCOMMS-24-00576-T

We would like to thank all the Reviewers for their highly constructive comments. We have conducted additional experiments and addressed all the points raised. We trust you'll agree that the careful revision process we have undertaken has substantially improved both the scientific rigor and implications of our study. We present point-by-point responses to each of the Reviewer comments (below). We start by listing the major experiments undertaken during our revision process.

- 1) We conducted additional *in vitro* experiments using INVI DNA Transfection Reagent to explore the long-term dynamics of light-responsive transcriptional regulation by the PTRC system (new **Supplementary Fig. 10**), demonstrating the sustained transcriptional regulation for over 6 days.
- 2) We conducted qPCR experiments to assess the duration required to completely reverse the transcriptional activation effect of the PTRC system (new **Fig. 3h**), demonstrating an immediate suppression of transcriptional activation upon exposure to red light, with the transcriptional deactivation reaching a stable state within approximately 5 hours.
- 3) We conducted further qPCR experiments to assess the capacity of the PTRC system for reversible and repeated operation (new **Fig. 3g**), demonstrating fully reversible transcriptional regulation kinetics mediated by the PTRC system across multiple cycles.
- 4) We conducted additional qPCR experiments to evaluate the time-dependent transcriptional regulatory activity of the PTRC_{CIP} and PTRC_{DL} systems, comparing their red light-induced kinetic profiles to their respective non-induced baselines (new **Fig. 5f, g** and **Fig. 6e, f**). The results demonstrated that red light-induced transcriptional deactivation of PTRC_{CIP} and PTRC_{DL} systems is significantly faster compared to the natural dissociation processes mediated by small molecule degradation or dark reversion.

Reviewer #1 (Remarks to the Author):

The author's manuscript presents an approach in synthetic biology for precise control of gene expression through synthetic transcription factors (sTFs). The authors validated the transactivator functions of two plant-derived phytochrome chaperone proteins (FHY1 and FHL) in mammalian cells and mice, forming a photosensing transcriptional regulation complex (PTRC). This complex can toggle between active and inactive states through exposure to red or far-red light. Leveraging this capability, the author constructed a light-switchable platform for orthogonal, modular, and tunable control of gene transcription, which successfully modulated both exogenous and endogenous genes in multiple mammalian cells and mice. This manuscript is original and provides a promising tool for synthetic biology and biomedical engineering. A few comments for this article are listed below:

Thank you very much for your highly positive comments.

1) Line 60-63: The description in this section is unclear and confusing in conveying the intended viewpoint. Please check it carefully and provide more comprehensive references to synthesize the current research gap.

Reply:

Thanks for the constructive comment. We have modified these sentences to clarify our intended viewpoint and more accurately reflect the current research landscape. Specifically, we have elaborated on the limitations of the control mechanisms in sTFs, emphasizing how their functionality is often constrained by the specificity of the DBD used. We highlighted the case of tetracycline-controlled sTFs, where the reliance on TetR as the DBD restricts their adaptability and compatibility with diverse gene regulatory networks. Please see **page 3** in the revised manuscript and below:

“However, the control mechanisms of these synthetic transcription factors (sTFs) are intrinsically linked to their specific DNA-binding domains (DBDs), which are often inflexible. For instance, in tetracycline-controlled sTFs, the DBD is specifically confined to the tetracycline-responsive protein TetR16. Changing the DBD to a non-TetR variant removes the tetracycline-mediated control, thus reducing their adaptability and compatibility with other systems.”

2) Line 84-85: Please supplement this section with references or list relevant works.

Reply:

Thank you for your suggestion. We have now included relevant citations to support the statement. Please see **page 4** in the revised manuscript and below:

“However, there is a notable shortage of plant-derived TFs effectively utilized in regulating transcription in mammalian systems²⁸.”

[28] Bhatt, B., García-Díaz, P. & Foight, G.W. Synthetic transcription factor engineering for cell and gene therapy. *Trends Biotechnol* (2023).

3) Line 128-130: The statement mentioning that FHY1 and FHL exhibited greater activation of SEAP expression than VP16 contradicts the results presented in Figure 1b. Please check it for accuracy.

Reply:

Thanks for the constructive comment. Upon reviewing the data, we found that FHL indeed demonstrated enhanced activation of SEAP expression compared to VP16 when fused with DBDs including TetR, CbaR, and VanR. However, FHY1 showed comparable or slightly less activation of SEAP expression relative to VP16. We have revised the statement to accurately reflect these findings and avoid any contradiction with the results. Please see **page 6** in the revised manuscript and below:

“FHY1 exhibited activation of SEAP expression that was comparable to, or slightly lower than, that induced by the positive control transactivator VP16, derived from herpes simplex virus⁵. In contrast, FHL when fused with TetR, CbaR, and VanR, showed significantly enhanced activation of SEAP expression compared to VP16.”

4) Please indicate in the legend of Figure 1h and 1i the specific regions removed in each truncation variant.

Reply:

Thank you for your valuable suggestion. To enhance the clarity and understanding of Figure 1h and 1i (revised Figure 1i and 1j), we have incorporated detailed schematics into Figure 1 (**Response Document Figure 1**). These schematics clearly delineate the specific regions removed in each truncation variant of FHL and FHY1. Please see **Fig. 1g, h** in the revised manuscript and below:

Response Document Figure 1 (see **Fig. 1g, h** in the revised manuscript): **g, h** Schematic of different truncation variants of FHL (**g**) and FHY1 (**h**), with different regions differentiated by color. The amino acid positions of the truncation sites are indicated at the top, and the amino acid regions for each truncation of FHL/FHY1 are detailed at the bottom. FL denotes full length. Mutants highlighted with a red box are referred to as Δ FHL/ Δ FHY1 in subsequent text.

5) Line 401-403: please check carefully whether this is the first article published to use FHL/FHY1 as transactivators in mammalian cells.

Reply:

Thank you for your reminder. After conducting a thorough review of the literature, we confirm that, to the best of our knowledge, our work represents the first instance of utilizing FHL/FHY1 as transactivators for synthetic transcription factors to regulate target gene expression in mammalian cells.

6) What is the long-term behaviour of PTRC system? How long will it take to completely remove the transcriptional activation effect?

Reply:

Given that the sTFs within the PTRC system are constitutively expressed, we hypothesized that transcriptional activation in non-illuminated cells would persist until either plasmid degradation or dilution due to cell division. To investigate this hypothesis and to further explore the long-term dynamics of the PTRC system, we conducted additional *in vitro* experiments. We introduced the PTRC system into HEK293T cells using INVI DNA Transfection Reagent, which is known to enhance plasmid stability within cells due to the electrostatic interactions between the cationic polymer and negatively charged plasmid DNA. Our observations revealed that FHL-induced SEAP production continued for over 6 days. Moreover, this activation was effectively suppressed by continuous exposure to red light (660 nm, 1 mW/cm²) (**Response Document Figure 2**). These results highlight the capability of the PTRC system for sustained transcriptional regulation. These new data have been incorporated into our manuscript's revised Supplemental Information (**Supplementary Fig. 10**). Please see **page 12** in the revised manuscript and below:

“To investigate the long-term dynamics of light-responsive transcriptional regulation, we delivered the PTRC transcriptional regulation system into HEK293T cells using INVI DNA Transfection Reagent to enhance plasmid stability⁴⁰. Continuous exposure to red light (660 nm, 1 mW/cm²) for more than 6 days effectively suppressed FHL-induced SEAP production (**Supplementary Fig. 10**), illustrating the system's capability for sustained transcriptional control.”

Response Document Figure 2 (see Supplementary Fig. 10 in the revised manuscript): The long-term dynamics of light-responsive transcriptional regulation by the PTRC transcriptional regulation system. The PTRC transcriptional regulation system were introduced into HEK293T cells using INVI DNA Transfection Reagent. Following transfection, the cells were kept in darkness or exposure to red light (660 nm, 1 mW/cm²). SEAP production was quantified daily. Data are presented as means ± s.d.; $n = 3$ independent experiments, with 3 technical replicates for each. Statistical comparisons were performed by two-way ANOVA; **** $p < 0.0001$.

To determine the time required to completely remove the transcriptional activation effect, we conducted experiments using HEK293T cells transfected with the PTRC system along with a SEAP reporter. Initially, these cells were kept in darkness for 12 hours to induce transcriptional activation. Subsequently, they were either exposed to red light (660 nm, 1 mW/cm²) or maintained in constant darkness. The mRNA levels of *SEAP* were minored hourly by qPCR. The results indicated an immediate suppression of transcriptional activation upon exposure to red light, with transcriptional deactivation stabilizing within approximately 5 hours (**Response Document Figure 3**). These new data have been incorporated into our revised manuscript (**Fig. 3h**). Please see **page 12** in the revised manuscript and below:

“To determine the time needed to fully reverse transcriptional activation in transfected HEK293T cells, we first induced activation by keeping the cells in darkness for 12 hours. Subsequently, the cells were either exposed to red light (660 nm, 1 mW/cm²) or kept in continuous darkness. Hourly measurements of SEAP mRNA levels showed that transcriptional activation was immediately suppressed upon exposure to red light, with transcriptional deactivation stabilizing within about 5 hours (**Fig. 3h**).”

Response Document Figure 3 (see Fig. 3h in the revised manuscript): Dynamics of red light-induced transcriptional deactivation. Transfected cells were first maintained in darkness for 12 hours and then either exposed to red light (660 nm, 1 mW/cm²) or kept in continuous darkness. The relative mRNA expression levels of *SEAP* were quantified using qPCR at intervals of 0, 1, 2, 3, 4, 5, 6, and 12 hours following the onset of red-light exposure.

7). Results showed in figure 3 indicated that this PTRC transcriptional regulation system is reversible by altering the illumination conditions. Is this reusable for multiple cycles? Please provide more details and discuss it.

Reply:

Thank you for your insightful suggestion. We have conducted further experiments to assess the system's capacity for reversibility and reuse over multiple cycles. Specifically, we transfected HEK293T cells with the PTRC system and subjected them to alternating periods of red-light exposure and darkness/far-red light, monitoring the mRNA levels of *SEAP* by qPCR. Our findings demonstrated that the PTRC system can indeed facilitate fully reversible transcriptional regulation kinetics across multiple cycles (**Response Document Figure 4**). These new data have been incorporated into our revised manuscript (**Fig. 3g**). Please see **page 12** and below:

“Moreover, the PTRC system demonstrated fully reversible kinetics over several cycles (**Fig. 3g**), highlighting its robustness and versatility for dynamic and precise gene expression control across various applications.”

g
Response Document Figure 4 (see Fig. 3g in the revised manuscript): Reversibility of PTRC-mediated transcriptional activation. Cells transfected with the PTRC system were either kept in darkness or exposed to red light (660 nm, 1 mW/cm²) or far-red light (730 nm, 1 mW/cm²), as indicated. The relative mRNA levels of *SEAP* were quantified by qPCR prior to any changes in the treatment conditions.

Reviewer #2 (Remarks to the Author):

In the presented manuscript the authors engineered synthetic transcription factors (sTFs) using the two plant-derived phytochrome chaperones FHY1 and FHL as transactivators. After investigating the homo- and heterodimerization capacity of the chaperones, DNA regions mandatory for proper protein functionality were identified. The authors then used the heterodimerization of the two chaperones to establish a split-Cre recombinase system, showing enhanced activity over the already known DocS/Coh2 approach. Afterwards, through reversible binding of PhyA and the sTF, formation of a photosensing transcriptional regulation complex was investigated and it was demonstrated, that the complex can be controlled by red, far-red light exposure. Finally, the authors presented integration of this system in different already established tools, like the CRISPRa system or small molecule- or blue light inducible modules. All in all, the presented approaches enabled reversible control of transcriptional regulation in vitro and in vivo.

In general, the study is presented in a comprehensive way and the claims made are supported by the data presented. Sometimes the inconsistency when displaying the fold-change or the significance in the same graph made it hard to interpret the given results and compare the individual datasets.

As the manuscript presents tools on protein-protein interaction as well as transcriptional regulation it can be of interest to a broad community in the fields of protein biology and transcriptional regulation, working in vitro and in vivo.

Even though the general concepts of the methods presented in the study already exist, the improved functionality and multiple combination possibilities shown for the tools generated, highlight their practicability in synthetic biology and biomedical engineering.

Thank you very much for your highly positive comments and we appreciate your interest in our work.

In order to better understand the manuscript and the presented data, it is suggested to address the following points.

Major

- As the speed of the system is mentioned several times as an advantage, it might be of interest to show some kinetic graphs, presenting and comparing the actual time-dependent activity. Optimally, a comparison is performed to previously published systems in order to evaluate the performance of the present approach compared to the state of the art.

Reply:

Thank you for your valuable suggestion. We have conducted additional experiments to assess the time-dependent activity of our systems, specifically focusing on the small molecule- and red light-induced transcriptional regulation (PTRC_{CRP}) system

and the dual-wavelength light-controlled transcriptional regulation (PTRC_{DL}) system. By comparing the kinetic profiles of these systems to their respective non-induced baselines, we observed that the red light-induced deactivation of transcriptional activation, which is initially triggered by small molecules or blue light, occurs at a significantly faster rate than natural dissociation processes mediated by small molecule degradation or dark reversion (**Response Document Figures 5 and 6**). This rapid response to red light highlights the efficiency and practical value of our systems, particularly for *in vivo* applications where quick removal of small molecules may not be feasible. These new data have been incorporated into our revised manuscript (**Fig. 5f, g and Fig. 6e, f**). Please see **pages 15 and 17** and below:

“Moreover, the red light-induced transcriptional deactivation mediated by the PTRC_{CIP} system occurred significantly faster than natural dissociation processes, which are often mediated by small molecule degradation (**Fig. 5f, g**). This finding indicates the potential of the PTRC_{CIP} system for *in vivo* applications, particularly in scenarios where the rapid removal of small molecules is not practical.”

“Compared to the natural dissociation processes mediated by dark reversion, the red light-induced transcriptional deactivation by the PTRC_{DL} system occurred significantly faster (**Fig. 6e, f**). This highlights the efficiency and potential of the PTRC_{DL} system for rapid control of gene expression.”

Response Document Figure 5 (see Fig. 5f, g in the revised manuscript): Dynamics of PTRC_{CIP} system. **f** Schematic representation of the timeline for PTRC_{CIP}-ABA-mediated transcriptional activation. HEK293T cells transfected with plasmids encoding the PTRC_{CIP}-ABA system were treated with 100 μM ABA in darkness for 12 hours before being separated into three groups. The first group was treated again with 100 μM ABA and kept in darkness (Dark + ABA). The second group was maintained in darkness without additional ABA treatment (Dark - ABA). The third group was exposed to red light (660 nm, 1 mW/cm²) without further ABA treatment (660 nm - ABA). The relative mRNA levels of *SEAP* were quantified using qPCR at 0, 1, 2, and 3 hours following the start of red-light exposure. **g** qPCR analysis of PTRC_{CIP}-ABA-mediated *SEAP* transcription as described in (**f**).

Response Document Figure 6 (see Fig. 6e, f in the revised manuscript): Dynamics of PTRC_{DL} system. e Schematic representation of the timeline for PTRC_{DL}-mediated gene expression. HEK293T cells transfected with plasmids encoding the PTRC_{DL} system were exposed to blue light (465 nm, 1 mW/cm²) for 12 hours before being separated into three groups. The first group of cells continued to be exposed to blue light (465 nm - 465 nm), the second group was maintained in darkness (465 nm - Dark), and the third group was exposed to red light (465 nm - 660 nm). The relative mRNA levels of *SEAP* were quantified using qPCR at 0, 1, 2, and 3 hours following the initiation of red-light exposure. f qPCR analysis of PTRC_{DL}-mediated *SEAP* transcription as described in (e).

- A key feature of light-controlled systems is the reversibility. It is recommended to experimentally demonstrate the kinetics of the reversibility of the system.

Reply:

Thank you for your insightful suggestion. We have conducted further experiments to assess the kinetics of the reversibility of the system. Specifically, we transfected HEK293T cells with the PTRC system and subjected them to alternating periods of red-light exposure and darkness/far-red light, monitoring the mRNA levels of *SEAP* by qPCR. Our findings demonstrated that the PTRC system can indeed facilitate fully reversible transcriptional regulation kinetics across multiple cycles (**Response Document Figure 7**). These new data have been incorporated into our revised manuscript (**Fig. 3g**). Please see **page 12** and below:

“Moreover, the PTRC system demonstrated fully reversible kinetics over several cycles (**Fig. 3g**), highlighting its robustness and versatility for dynamic and precise gene expression control across various applications.”

g

Response Document Figure 7 (see Fig. 3g in the revised manuscript): Reversibility of PTRC-mediated transcriptional activation. Cells transfected with the PTRC system were either kept in darkness or exposed to red light (660 nm, 1 mW/cm²) or far-red light (730 nm, 1 mW/cm²), as indicated. The relative mRNA levels of *SEAP* were quantified by qPCR prior to any changes in the treatment conditions.

Minor

- Easy comparison of conditions is not possible if either significance or fold-change is given for a data set in the same graph, stick to one or present both (eg. 3c (statement not supported by data?), 9a, b).

Reply:

Thank you for your valuable suggestion. We have revised our presentation of the data and provided significance analysis for all datasets to ensure consistency and clarity. Please refer to the updated figures in the revised manuscript.

- Addition of the number of technical replicates performed to the figures.

Reply:

Thank you for your insightful recommendation. We have updated the figure legends in the revised manuscript to include the number of technical replicates performed for each experiment. Please refer to the updated figure legends in the revised manuscript.

- Figure 1a and S1a are identical.

Reply:

Thank you for your comment. We have revised Figure 1a and Supplementary Fig.1a to ensure they distinctly represent the information. Specifically, we have included detailed representations of the synthetic genetic elements involved, which helps in differentiating between the figures and enhances their clarity. Please see the revised figures in the manuscript and below:

a

Response Document Figure 8 (see Fig. 1a in the revised manuscript): Schematic of the design for screening potential transactivators. The DNA binding domain (DBD: Gal4 or TetR) is fused with candidate transactivators to create synthetic TF (sTF), which translocates into the nucleus and binds to the synthetic specific promoter to initiate target gene expression.

a

Response Document Figure 9 (see Supplementary Fig. 1a in the revised manuscript): Schematic of the design for screening potential transactivators. The DBD (CbaR or VanR) is combined with the candidate transactivator (PIF3, FHY1, FHL, or VP16) to generate a synthetic TF (sTF), which translocates into the nucleus and binds to the synthetic specific promoter to initiate the target gene expression.

- Figure 1 statistical test used not indicated.

Reply:

Thank you for your comment. We have updated the figure legends in the revised manuscript to include the statistical test used. Please see **page 41** in the revised manuscript and below:

“Statistical comparisons were performed by one-way ANOVA; NS, not significant, *** $p < 0.001$, **** $p < 0.0001$.”

- Figure 2m not really visible, maybe enhance quality or zoom in.

Reply:

Thank you for your valuable suggestion. We have improved the visibility of Figure 2m by zooming in on the key areas of interest. Please see revised **Fig. 2m** and below:

Response Document Figure 10 (see Fig. 2m in the revised manuscript): Representative fluorescence images of the liver sections of the transgenic Cre-tdTomato reporter mice shown in (i). Scale bar, 50 μ m.

- In Figure S4b, the meaning of the red box is not indicated and from line 203 on, a Δ FHL/ Δ FHY1 version is mentioned, that is not clearly defined anywhere (equivalent to red box/ Δ 5?), also it might be nice to add S4b to the main Figures, to improve understanding.

Reply:

Thank you for your valuable suggestion. The Δ FHL/ Δ FHY1 version mentioned in the manuscript corresponds to the variant indicated by the red box in Figure S4b, which we refer to as Δ 5. We apologize for any confusion this may have caused. Based on your suggestion, we have now incorporated the schematic that was initially in Figure S4b into **Fig. 1g, h**. Please see **page 9** in the revised manuscript and below:

“By contrast, deleting the C-terminus (117-201 aa) resulted in significantly increased split-Cre recombinase activity through dimerization of the Δ FHL and/or Δ FHY1 variants (denoted as FHL Δ 5/FHY1 Δ 5 in Fig. 1g and h), reaching three to five-fold higher activity than that detected with full-length protein.”

Response Document Figure 11 (see Fig. 1g, h in the revised manuscript): **g, h** Schematic of different truncation variants of FHL (**g**) and FHY1 (**h**), with different regions differentiated by

color. The amino acid positions of the truncation sites are indicated at the top, and the amino acid regions for each truncation of FHL/FHY1 are detailed at the bottom. FL denotes full length. Mutants highlighted with a red box are referred to as Δ FHL/ Δ FHY1 in subsequent text.

- Line 199 typo (split-Cre), also the sentence is not really clear to me.

Reply:

Thank you for your careful review. The phrase “a half component of spit-Cre recombinase system” refers to either CreN or CreC fused with various truncations of FHY1/FHL. To improve clarify, we have revised the sentence to more accurately describe the experimental design. Please see **page 9** in the revised manuscript and below:

“By contrast, deleting the C-terminus (117-201 aa) resulted in significantly increased split-Cre recombinase activity through dimerization of the Δ FHL and/or Δ FHY1 variants (denoted as FHL Δ 5/FHY1 Δ 5 in Fig. 1g and h), reaching three to five-fold higher activity than that detected with full-length protein. Notably, minimal SEAP production was observed when either CreN or CreC fused with various Δ FHL/ Δ FHY1 variants (**Fig. 2b**).”

- Figure S8b, there is already a significant drop stated without any plasmid added? Supposed to be 10 ng instead of 0?

Reply:

Thank you for pointing out this mistake. It appears there was an error in labeling the x-axis. We have corrected this issue in the revised manuscript. Please see new **Supplementary Fig. 8** in the revised manuscript and below:

Response Document Figure 12 (see Supplementary Fig. 8b in the revised manuscript): Quantification of PTRC-mediated gene expression with varying amounts of plasmids encoding Δ PhyA-2xZIM3. Transfected HEK293T cells (6×10^4) were illuminated using red

light (660 nm, 1 mW/cm²) or maintained in darkness for 24 hours before quantifying SEAP production. Control cells were transfected with plasmids encoding Gal4-FHL, the corresponding reporter, and pcDNA3.1(+).

- Figure 6, title: PTRCDL instead of PSTDL?

Reply:

Thank you for catching that mistake in the title of Figure 6. We apologize for the confusion and have corrected it to “PTRC_{DL}” in the revised manuscript. Please see **page 53** in the revised manuscript and below:

“Fig. 6 A dual-wavelength light-controlled transcriptional regulation (PTRC_{DL}) system for controllable transcriptional activation and deactivation in mice.”

Reviewers' Comments:

Reviewer #1:

Remarks to the Author:

I have no more comments.

Reviewer #2:

Remarks to the Author:

The authors have performed comprehensive additional experiments and further reworked the manuscript. They have adequately addressed the reviewers' concerns. The added data convincingly demonstrate the functionality and applicability of the system.

Manuscript number: NCOMMS-24-00576A

Point-by-point responses to referees' comments:

REVIEWERS' COMMENTS

Reviewer #1 (Remarks to the Author):

I have no more comments.

Reply:

We are happy that this reviewer is satisfied with our revision work. Thank you.

Reviewer #2 (Remarks to the Author):

The authors have performed comprehensive additional experiments and further reworked the manuscript. They have adequately addressed the reviewers' concerns. The added data convincingly demonstrate the functionality and applicability of the system.

Reply:

Thanks for your positive feedback on our revised manuscript. We are pleased to hear that we have satisfactorily addressed your concerns.